



# Storm Boris (2024) in the current and future climate: a dynamics-centered contextualization, and some lessons learnt

Jacopo Riboldi[1,*], Robin Noyelle[1,*], Ellina Agayar[1], Hanin Binder[1], Marc Federer[1],
Katharina Hartmuth[1], Michael Sprenger[1], Iris Thurnherr[1], and Selvakumar Vishnupriya[1]

[1]Institute for Atmospheric and Climate Science, ETH Zurich, Zurich, Switzerland
[*]*These authors contributed equally to this work*

**Correspondence:** jacopo.riboldi@env.ethz.ch

**Abstract.**

The response of mean and extreme precipitation to anthropogenic global warming stems both from warming of the troposphere and dynamical changes in the large-scale circulation, especially upward motions. The interaction between these two components complicates future projections, and makes the attribution of extreme precipitation events challenging, both using
conditional (e.g., analog-based) and unconditional (e.g., extreme value theory-based) methods. In this study we reflect upon this problem and propose some possible solutions to tackle it starting from the case study of Storm Boris, that led to major floods over central Europe in mid-September 2024. The first step is the identification of key circulation features associated with the event, whose representation is deemed crucial to obtain realistic analogs: the presence of a slow-moving, upper-level potential vorticity (PV) cutoff, the peculiar track of the surface cyclone associated with Boris, and the presence of anomalously
strong forcing for ascent. Circulation analogs of Boris are then identified in a large ensemble of present-day and future-climate simulations with the CESM1 model, to understand how Boris-like storms will change in an end-of-the-century high warming scenario. We find that the combined use of upper-level PV and of a surface cyclone identification algorithm substantially improves the quality of the analogs, both in terms of the large-scale flow pattern and the precipitation associated with the cyclone. Analogs of Boris restricted to the same season in a warmer climate feature on average less precipitation, due to an
overall weakening of upper-level-driven ascent over Europe. However, analogs of Boris not restricted to the same season show a seasonality shift, becoming less frequent ad the end of the warm season and more frequent in the shoulder seasons – when the dynamical and thermal conditions of September in the present-day climate can be recovered again –, and exhibit an increase in mean precipitation in the warmer climate. The results obtained from the analog-based approach are then compared with an unconditional, statistics-based approach focusing only on the yearly maximum of precipitation, recovering the expected inten-
sification of extreme precipitation in a warmer climate – at the price, however, of considering events that do not necessarily have the same dynamics as Storm Boris. The sensitivity of attribution outcomes with respect to implicit and explicit methodological choices is discussed in detail. The systematic comparison of different approaches, the two-step methodology to obtain more reliable analogs of heavy precipitation events, and the focus on process understanding are key ingredients of this study, with general implications for investigating the role of climate change for specific weather extremes.





## 1 Introduction

Together with heat extremes, daily and multi-day precipitation extremes are globally expected to become more frequent and impactful with global warming (Seneviratne et al., 2021). The main reason behind this trend is the increased availability of atmospheric moisture with higher temperatures (Clausius-Clapeyron scaling; e.g., Hartmann et al., 2013; Ban et al., 2015; Fischer and Knutti, 2016; Kim et al., 2022). This tendency, however, might be regionally counteracted by changes in atmospheric dynamics (O'Gorman and Schneider, 2009; Pfahl et al., 2017a; Allan et al., 2020; Dallan et al., 2024), which are generally less constrained than thermodynamical effects (Shepherd, 2014; Di Capua and Rahmstorf, 2023). Those competing signals make local assessments of future changes in precipitation extremes particularly challenging. The Euro-Mediterranean region is expected to be particularly affected by this negative feedback, with a dynamically-driven mean drying (Seager et al., 2014; D'Agostino and Lionello, 2020) and, at the same time, an increase in frequency and intensity of precipitation extremes (e.g., Rajczak et al., 2013). Changing dynamics can oppose the expected precipitation increase directly, via a weakening of atmospheric vertical motions (Pfahl et al., 2017a; Tandon et al., 2018), and indirectly, via changes in the event seasonality, which lead to the occurrence of the extremes in different periods of the year (Brönnimann et al., 2018; Marelle et al., 2018; Müller et al., 2024). The added layer of complexity provided by dynamics calls for a deeper understanding of atmospheric dynamical processes when placing extreme precipitation events in the context of the changing climate, i.e., in detection and attribution studies (Sarojini et al., 2016; Shepherd, 2016; Terray, 2021; Thompson et al., 2024). This piece of research focuses precisely on the complexity of attributing extreme precipitation events to anthropogenic global warming: it moves its steps from the case of Storm Boris (2024), a recent example of a cutoff-related, heavy precipitation event over central Europe, and studies how its characteristics change in a warmer climate as simulated by the medium-resolution climate model CESM1 large ensemble with more than a thousand years of data using an analog-based approach.

Storm Boris led to a record-shattering precipitation event: in the only five days between the 12th and 16th September 2024, the equivalent of 2 to 5 times the climatological precipitation totals for September fell over a broad region of central Europe, leading to widespread flooding that severely impacted local communities (Greilinger et al., 2024; Copernicus Climate Change Service, C3S). This extreme event was linked to the development of an extratropical cyclone, which was named 'Boris' by the Italian Meteorological Service. As other European heavy precipitation events (e.g., Grams et al., 2014; Ferreira, 2021; Tradowsky et al., 2023; Flaounas et al., 2024) its cyclogenesis was preceded by Rossby wave breaking over central Europe, that led to the formation of an upper-level PV cutoff over the central Mediterranean Sea. Wave breaking events tend to be associated with enhanced low-level poleward advection of warm, moist air ahead (i.e., to the east) of the incipient PV cutoff and with strong ascending motions, both processes promoting the occurrence of extreme precipitation (Llasat et al., 2007; Moore et al., 2019; de Vries, 2021). Furthermore, the circulation anomalies associated with breaking Rossby waves are strong enough to locally oppose the background westerly flow, leading to a persistent flow pattern that generally increases the multi-day accumulation of precipitation (e.g., Lenggenhager et al., 2019; Lenggenhager and Martius, 2019; Kautz et al., 2022).

While the upper-level flow evolution was important in setting the stage for the event, there were also interesting low-level features at play. Boris started as a Mediterranean surface cyclone that crossed Italy and then moved over the Balkans, following





a track reminiscent of the so-called 'Vb cyclones' (Van Bebber, 1891; Hofstätter and Chimani, 2012). The occurrence of such

cyclones had been related to extreme precipitation events over Central Europe and the northern Alpine region (e.g., Ulbrich et al., 2003; Messmer et al., 2015; Hofstätter et al., 2016). Storm Boris also occurred in a period of anomalously high sea surface temperatures (SSTs) in the Mediterranean, a pre-conditioning that can lead to enhanced precipitation from Vb cyclones (Messmer et al., 2017).

The rapid attribution studies performed so far suggested that anthropogenic global warming likely increased the precipitation

totals related to Storm Boris. Athanase et al. (2024) used a nudging method to impose the same large-scale, upper-tropospheric dynamics as during the event in a present-day climate model simulation and found an increase of precipitation with respect to a pre-industrial climate of around 9% in the impacted region, driven by the increase in vertical motion and available water vapor. The changes with regards to a warmer (+4 K) climate than today are more subtle, with only a small increase in precipitation (+2%) and some regions even seeing less precipitation. Likewise, using an extreme value analysis over a large number of

models and observational data sets, Kimutai et al. (2024) found an increase of the precipitation intensity of the event compared to a pre-industrial climate by 7%.

Using large-scale circulation analogs in the observational record in the context of the ClimaMeter rapid attribution framework, Faranda et al. (2024) compared Boris-like cyclones in the present climate (obtained from analogs in the period 2001-2023) and in a counterfactual climate (obtained from analogs in the period 1979-2001). They also observed an increase in

precipitation, and related it to a deepening of Boris-like storms in the present climate with respect to the counterfactual. However, as discussed in Faranda et al. (2024), confidence in the results is partially limited by the relatively small number of analogs of Boris found in the ERA5 reanalysis data set, a common issue when studying unusual extreme events. Performing analog-based studies of heavy precipitation events related to upper-level PV cutoffs is particularly challenging, as small differences in the PV distribution can result in notable shifts in the regions affected by heavy precipitation (as noticed by Fehlmann and

Quadri, 2000; Fehlmann et al., 2000; Schlemmer et al., 2010; Thompson et al., 2024). This observation further underscores the importance of correctly representing the dynamical setup of extreme precipitation events when performing detection and attribution studies, an aim that can be achieved through improved large-scale circulation analogs.

In this study, we try to harness knowledge about the dynamics of Storm Boris to obtain a dynamics-informed, analog-based assessment of how similar events will look like in a future, warmer climate. While getting to such a result, we will reflect upon

how the knowledge of atmospheric dynamics can be effectively implemented in analog-based approaches to study the effects of climate change on extreme precipitation events, and contextualize how different methodological choices affect the outcomes of attribution. The large ensemble simulations, the attribution approach, and the diagnostics used to investigate the dynamics of the event are introduced in Sect. 2. The analysis of future changes in Storm Boris is preceded by a detailed description of the dynamics of the event (Sect. 3), which results in the identification of a small set of salient dynamical features deemed

particularly relevant for the unfolding of the extreme event. In Sect. 4.1, the selection of the analogs is then performed in a way that ensures an appropriate representation of these salient dynamical features. A conditional, analog-based attribution of Storm Boris is then performed: its results are contextualized with respect to methodological choices, and compared with an unconditional attribution method in the rest of Sect. 4. As we believe that the explicit consideration of dynamics, and the



dependence of attribution outcomes on the chosen approach, might be important for attribution studies of extreme precipitation events in general, we discuss some more generic insights in Sect. 5, before presenting the overall conclusions of the study and some lessons learnt in Sect. 6.

## 2 Data and methods

### 2.1 Data

Meteorological data are obtained from 6-hourly analysis data of the Integrated Forecasting System (IFS, version Cycle 48r1) model of the European Centre for Medium-Range Weather Forecasts (ECMWF), interpolated to a horizontal resolution of $0.5° \times 0.5°$ degrees. The 5-day accumulated precipitation fields stem from the global atmospheric reanalysis dataset of the European Centre for Medium-Range Weather Forecasts ERA5 (ECMWF Hersbach et al., 2020). To obtain a more accurate estimate of rainfall during the event, we also use the Late Run IMERG V07 product from the Integrated Multi-satellitE Retrievals for GPM (IMERG) dataset (Huffman et al., 2024), at half-hourly intervals and $0.1°$ resolution in space, which aligns fairly well with station observations.

The contextualization of Boris with respect to climate change is based on a large ensemble of the Community Earth System Model version 1 (CESM1; Hurrell et al., 2013). Combining the original 35 members of the CESM1 large ensemble (Kay et al., 2015) with other "micro-ensemble" simulations initialized from the first and the second member of the large ensemble (as detailed in Fischer et al., 2013; Röthlisberger et al., 2020), we obtain two sets of 105 10-year members for the periods 1990–1999 (historical) and 2090-2099 (future, run under a Representative Concentration Pathway 8.5 forcing scenario). Model output has a horizontal resolution of $1.25°$ of longitude by $\sim0.9°$ of latitude, 6-hourly temporal resolution, and 30 vertical levels, allowing to reliably perform advanced dynamical analyses such as the computation of air parcel trajectories, and of quantities involving vertical gradients, such as PV.

### 2.2 Analog-based attribution

Attribution of Storm Boris to climate change is performed through the analysis of flow analogs in the available large ensemble of the CESM1 climate model. Flow analogs of an event with atmospheric circulation $\mathcal{C}_e$ are days $d$ when the atmospheric circulation $\mathcal{C}_d$ is sufficiently similar to $\mathcal{C}_e$ according to a certain metric. These analogs give an empirical estimation of the distribution of variables of interest $X$ conditional on the atmospheric circulation during the event: $\rho(X|\mathcal{C}_e)$. One can then compare the evolution of properties of the conditional distributions in the model between the two periods ($F = 0$ and $F = 1$), for example the mean change in a variable $X$: $\Delta X_{\mathcal{C}_e} := \mathbb{E}(X|\mathcal{C}_e, F = 1) - \mathbb{E}(X|\mathcal{C}_e, F = 0)$. $\Delta X_{\mathcal{C}_e}$ informs how the expected value of $X$ is changing between the two periods — and therefore with increased anthropogenic forcings in our case — conditional on the atmospheric circulation being similar as the one observed during the event. Contrary to unconditional attribution methods, this method ensures that the atmospheric dynamics of events used to compare to the event observed are similar in both periods and, therefore, that dynamically similar events are compared.



In this study, analogs are defined based on the euclidean distance between $\mathcal{C}_d$ and $\mathcal{C}_e$ over a broad domain encompassing Europe (32°-70°N and 5°W-40°E, see Fig. 6f for the box over which the analogs are computed). While other studies employing the flow-analogs attribution method used either geopotential height at 500 hPa (Z500) or sea-level pressure (SLP) to condition on the large scale circulation $\mathcal{C}_e$ of the event (Terray, 2021; Faranda et al., 2022; Noyelle et al., 2023; Thompson et al., 2024) here we prefer to use the 250 hPa PV field to obtain an improved representation of the dynamics of Storm Boris. The reasoning
behind this choice will be made explicit in the following sections, but it is worth remembering here that PV is able to provide a powerful and compact representation of the atmospheric flow (as shown by, e.g., Hoskins et al., 1985). We thus compute analogs of the event based on the daily mean 250 hPa PV field on the 14th of September. Furthermore, for each daily mean PV field, we subtract the spatial mean of the PV field over the domain before computing the euclidean distances: this is done to select analogs with similar PV gradients to the event — rather than raw PV intensity, as this field is expected to change with
global warming for a fixed pressure level due to the expansion of the tropical belt (Turhal et al., 2024).

    Attribution is then performed by comparing circulation analogs in a *counterfactual* period of the model with limited anthropogenic forcings (1990–1999, referred to as "the present" in the following) and a *factual* period (2091–2100, referred to as "the future" in the following) with high anthropogenic forcings —and a global warming of 4.5 K compared to the counterfactual. In each period we look for analogs in the large ensemble of 105 members of the CESM1 model (Hurrell et al., 2013), which we
assume is equivalent to looking for analogs in a 1050-years stationary simulation. We first select the closest $n = 1050$ analogs in each period based on their euclidean distance with respect to the PV field of the event (again, after removing the domain-averaged PV), while imposing that each analog is separated by at least 5 days from a previously selected analog to ensure that they are not taken from the same event in the model. This number of analogs was chosen to have the equivalent of one analog per year in the model, although we do not impose that one analog per year has to be found. Two ways of selecting flow analogs
are considered: 1) "seasonal analogs", by restricting analog computation to the same season of the event (i.e., only the months of August, September, October), and 2) "yearly analogs", for which no restriction on months is imposed. Both approaches are compared to investigate potential seasonality changes between analogs in the factual and the counterfactual period, and to assess the impact of methodological choices on the outcome of the attribution assessment.

    For every metric computed on the analogs distribution, we test the significance of the difference between the two periods
by estimating a bootstrapped distribution with 1000 resampling with replacement over the analogs. We then compute the probability in this resampled distribution that the value 0 (or 1 when considering ratios) is exceeded/subceeded. This gives a bootstrapped p-value for significance at each grid point. We then use a false discovery rate test of 0.1 to take into account spatial correlations (Wilks, 2016) and flagged as significant the grid points that pass this test.

## 2.3   Dynamical diagnostics

**Warm conveyor belts**   Warm conveyor belts (WCBs) are identified in the operational ECMWF analyses and the CESM simulations following the method of Madonna et al. (2014). Forward trajectories are computed over a two-day period using the Lagrangian Analysis Tool (LAGRANTO; Wernli and Davies, 1997; Sprenger and Wernli, 2015). They are started every 6 h from an equidistant grid covering the lower troposphere (1050-790 hPa) of the Northern Hemisphere, with a





horizontal resolution of 40 km and a vertical resolution of 20 hPa. Those trajectories that ascend at least 600 hPa within
these two days are classified as WCB trajectories.

**Atmospheric blocking**  Blocking identification in the operational ECMWF analysis and CESM simulations follows the approach of Schwierz et al. (2004) and Croci-Maspoli et al. (2007). Regions with a vertically averaged negative PV anomaly (between 150 and 500 hPa) exceeding -0.7 PVU, relative to a 15-day running mean, are classified as blocks if the anomaly persists for at least five consecutive days. The 15-day running mean used to compute the anomalies, on the other hand, is derived from ERA5 reanalysis data. Since Storm Boris occurred in early fall — when PV anomalies tend to be weaker — a lower PV anomaly threshold of -0.7 PVU is applied (as in, e.g., Pfahl and Wernli, 2012).

**Cyclone tracking**  The track of Storm Boris was identified manually in the ECMWF analysis, matching minima in SLP and geopotential height at 850 hPa. This was necessary because the representation of orography in the ECWMF analysis introduces noise to the SLP field, making automatic tracking impractical. Identification and tracking of cyclones in CESM1 is based on the SLP field using the contour search algorithm developed by Wernli and Schwierz (2006) and refined by Sprenger et al. (2017). Each cyclone is characterized by a pressure minimum, which is followed over time to generate a cyclone track.

**Moisture source**  Moisture sources were calculated from ten-day backward air parcel trajectories initialized every 50 km within the target box shown in Fig 1, at pressure levels from 1000 to 500 hPa in 50 hPa increments. Trajectories were computed using LAGRANTO, applied to the 3D wind fields from ECMWF analysis data. Specific humidity, relative humidity, and boundary layer height were interpolated along each trajectory to compute the moisture source diagnostic following Sodemann et al. (2008). Moisture uptakes were identified as increases in specific humidity exceeding $0.01 \, \text{g kg}^{-1}$, and were weighted by moisture losses (larger than $-0.01 \, \text{g kg}^{-1}$) after uptake. Only trajectories that contributed to precipitation - defined as a decrease in specific humidity greater than $0.01 \, \text{g kg}^{-1}$ and a relative humidity above 80% in the final time step before arrival - were considered. Finally, each uptake was weighted by the final moisture loss of its trajectory. These weighted uptakes were then gridded onto a 0.5° global latitude-longitude grid. Gridded relative moisture uptakes were then scaled using surface precipitation data from IMERG for each time step in the target region. On average, this method explains $90 \pm 1\%$ of the precipitation in the target region between 12 and 16 September 2024. To assess the influence of anomalous Mediterranean SSTs on moisture uptake, the same setup was applied using ERA5 reanalysis data (Hersbach et al., 2020), interpolating daily SST values and their 30-year climatology (with respect to the 1994–2024 period) along the trajectories.

**Quasi-geostrophic forcing for ascent**  Quasi-geostrophic (QG) forcing of ascent and descent in the operational ECMWF analysis and the CESM simulations is calculated as in Besson et al. (2021) and Jin et al. (2024). Specifically, all fields are coarse-grained to a 1.5 x 1.5 latitude/longitude grid, and then the Q-vector and its divergence are determined as forcing of the vertical motion, thereby distinguishing between forcings from levels above and below 550 hPa. More details on





the Q-vector formulation of the QG theory can be found in Davies et al. (2024), and a technical documentation of the calculation method in Reinert (2009).

**Extreme precipitation objects** To investigate the association of analogs with the occurrence of extreme precipitation, we define 2-dimensional objects of extreme precipitation as follows. First, we determine a threshold at each grid point based on the 75th percentile of precipitation based on a monthly climatology. In a second step, grid points that exceed that threshold are defined as extreme precipitation objects described by a two-dimensional binary field that obtains a value of 1 at grid points within the object, and a value of 0 outside. Taking a temporal average from these binary fields for all analogs will result in the probability of the occurrence of extreme precipitation at each grid point.

## 2.4 Precipitation scaling

To investigate changes in the precipitation between the factual and counterfactual, we make use of the precipitation scaling proposed by O'Gorman and Schneider (2009). The intensity of precipitation extremes scales as:

$$P \sim -\{\omega \left.\frac{dq_{sat}}{dp}\right|_{\theta^*,T}\} := P_{sca} \tag{1}$$

where $P$ is a high percentile of the precipitation distribution, $\omega$ is the large-scale average over precipitation systems of vertical velocity in pressure coordinates, $\left.\frac{dq_{sat}}{dp}\right|_{\theta^*,T}$ is the moist-adiabatic derivative of saturation specific humidity conditional on the mean temperature when extreme precipitation occurs and $\{\cdot\}$ is a mass-weighted integral over the troposphere. In the following for simplicity of writing we drop the condition $\theta^*,T$ for the saturated specific humidity. Importantly, for the $\omega$ term we consider only updrafts, i.e. when $\omega < 0$. When $\omega$ is positive it will be considered as being equal to 0. Although this scaling is more adapted to intense precipitation events (O'Gorman and Schneider, 2009; Pfahl et al., 2017b), we use it here for every analog even if they do not display intense precipitation, and discuss the validity of this approximation in the results section.

This scaling allows to separate between thermodynamical — change in $\frac{dq_{sat}}{dp}$ via the Clausius-Clapeyron relationship — and dynamical — change in $\omega$ — contributions to extreme precipitations. In particular, this allows to decompose the evolution of future precipitation into those two components. We note $< \cdot >_k$ the analogs average in period $k$. Then for each analog:

$$P_{sca} = -\{(< \omega >_k - \omega')(< \frac{dq_{sat}}{dp} >_k - \frac{dq_{sat}}{dp}')\}, \tag{2}$$

where the primes describe the anomalies compared to the analogs average. This implies that the mean precipitation for the analogs can be separated into a contribution from the means and a contribution from the covariance:

$$< P_{sca} >_k = -\{< \omega >_k < \frac{dq_{sat}}{dp} >_k + < \omega' \frac{dq_{sat}}{dp}' >_k\}, \tag{3}$$

where by linearity we switched vertical and analogs averages. We then note $\Delta X := < X >_{2091-2100} - < X >_{1990-1999}$ the changes in the analogs average between the two periods for any variable $X$. The change in the analogs average scaling precipitation can thus be written as the sum of four terms:





$$\Delta P_{sca} = -\{\Delta\omega < \frac{dq_{sat}}{dp} >_{1990-1999} + <\omega>_{1990-1999} \Delta\frac{dq_{sat}}{dp} + \Delta\omega\Delta\frac{dq_{sat}}{dp} + \Delta\omega'\frac{dq_{sat}}{dp}' \}. \tag{4}$$

The first two terms represent respectively the change in scaling precipitation due to changes in the mean vertical wind speed and mean vertical saturated specific humidity. The latter can therefore be understood as a mean dynamical contribution while the former is the mean thermodynamical contribution due to the change in the vertical profile of saturated specific humidity. The third term corresponds to changes of both means at the same time and is likely of second-order importance compared to the first two. Finally the last term represents a change in the covariance between the two periods, i.e. whether anomalies of the vertical wind speed occur more at the same time as anomalies of the saturated specific humidity in the future compared to the present. This term need not a priori be small compared to the first two. The decomposition in Eq. 4 allows then to attribute changes in the scaling precipitation to different changes in atmospheric processes between the two periods.

## 3 Salient dynamical features of Storm Boris

### 3.1 Spatial and temporal extent of the event

We focus here on the extended, spatially coherent extreme precipitation event that unfolded over central-eastern Europe in the period between the 12th and the 16th of September 2024, and neglect the shorter-duration, more localized extreme rainfall events that impacted Romania and Italy during these days, respectively, preceding and following the main event (Ferrari et al., 2025). In that time span, locally more than 350 mm of precipitation were recorded between Austria, Germany and Czech Republic, leading to widespread flooding in the Danube river basin. Despite the systematically lower values, the distribution of precipitation captured by the ERA5 analysis dataset matches well the IMERG observational data set (Figs. 1a,b). In terms of timing, both data sets highlight the 48-h period between 00UTC of the 13th and the 15th of September 2024 as the peak of the event (Fig. 1c), and the analysis will focus on this time interval.

### 3.2 Large-scale flow evolution

Cyclone Boris originated from the Rossby wave breaking of a PV streamer (i.e., a filament of stratospheric air intruding in the troposphere, see Appenzeller and Davies, 1992) elongated between Norway and Italy, which was associated with unseasonably cold air masses over western and central Europe in the days preceding the event (Fig. 2a). Heavy precipitation started to affect central Europe already during the 13th of September, focusing at the north-eastern side of the cyclonically breaking PV streamer (Fig. 2b). Precipitation concentrated along a baroclinic zone between the pre-existing cold air, associated with the PV streamer to the west, and the warm air advected cyclonically around the streamer from the south-east and upgliding above the cold air mass. Ascent was particularly strong at this stage, with many air parcels matching the WCB criterion of 600 hPa uplift in 48 h in correspondence of the baroclinic zone (Fig. 2b). It is also worth noticing that the wave breaking happened in conjunction with an atmospheric blocking over the Barents Sea, a feature that likely hindered the propagation of the PV trough at high latitudes and, consequently, promoted its cyclonic breaking over southern Europe.



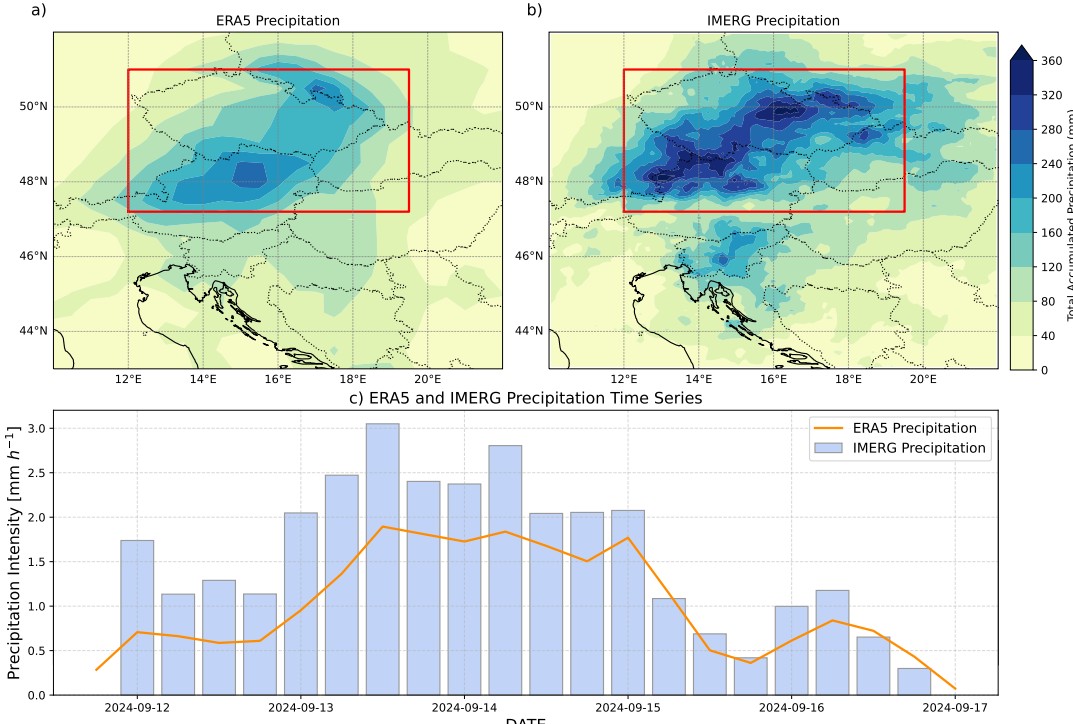

**Figure 1.** Precipitation totals in the 5-day period between 00UTC of the 12th of September and the 17th of September 2024, as estimated from (a) 6-hourly ERA5 analysis and (b) the IMERG satellite product. (c) Time series of precipitation intensity (in mm/h, averaged over 6 h intervals) over the considered impact region, marked by the red box in (a) and (b).

The streamer progressively filamented during the 13th of September, leading to the formation of a broad PV cutoff between Italy and the Balkans (Fig. 2c). Filamentation was accompanied by substantial WCB activity at *both* the western and eastern sides of the streamer, as indicated by the many WCB outflow trajectories located in its immediate vicinity (Figs. 2a,c). Precipitation intensity reached at this time a peak over Austria and Czech Republic with maxima located again at the northern edge of the PV cutoff, along the quasi-stationary baroclinic zone, and in correspondence of the Alps over western Austria —

indicating an orographic contribution (Fig. 2d). We also notice that WCB activity is now confined at the eastern edge of the PV cutoff, away from the heaviest precipitation falling over the impacted region: this observation points to the contribution of other processes to heavy precipitation during the PV cutoff stage, such as isentropic upgliding and orographic enhancement.

    The final part of the event was characterized by more moderate yet persistent precipitation during the 15th and the 16th of September, as the cutoff and the surface cyclone stalled and started to retrograde westward towards northern Italy (Figs. 2e,f).

The precipitation swath follows again the region of strong $\theta_e$ between Poland and Austria, indicating the persistence of isentropic upgliding along the now weaker, north-south oriented baroclinic zone. The persistence of the flow pattern can be ex-





plained by the presence of high-latitude blocking over northern Europe broadly promoting easterlies over Europe that first halt the eastward propagation of the PV cutoff and then support its retrogression (see Fig. S1).

### 3.3 Low-level flow evolution

The surface cyclone associated with Boris originated close to the Gulf of Genoa between the 11th and the 12th of September (Fig. 3a), at the southern edge of a broad upper-level PV streamer located over central Europe. As the streamer proceeded eastward and started to break cyclonically, the center of the cyclone was stirred across Italy towards the Carpathians (Fig. 3b): we verified that such a track fulfills the definition of Vb cyclone introduced by Hofstätter and Chimani (2012) and Messmer et al. (2015). We remind here that the track shown in Figs. 3a,c,e results from an ad-hoc, manually performed tracking of Boris'

SLP minimum and geopotential at 850 hPa. The automated tracking algorithm, on the other hand, split Boris' track between two separate lows despite visual evidence of a single, broad, stationary cyclone.

Analysis of integrated vapor transport (IVT) highlights patterns of moisture advection first to the south of the PV streamer (Fig. 3a) and then around the cutoff cyclone towards the impacted region (Figs. 3c,e). However, since IVT provides only a snapshot of the moisture transport at a given time, we complemented this analysis by a trajectory-based moisture source

diagnostic focused on the region impacted by the heaviest precipitation. The diagnostic indicates that, in the first part of the event, most of the moisture contributing to precipitation originated from central to western Europe (40%), the western and central Mediterranean (18%) and the North Atlantic (23%) (Fig. 3d). As the event unfolded, however, contributions from eastern Europe (40%) and the Black and Caspian Sea (10%) became increasingly important (Fig. 3e). The detailed partition of moisture uptakes among the different regions is displayed in Fig. S2. Notably, over 90% of the extended Mediterranean

contribution (i.e., conflating together the whole Mediterranean and Black Sea) during the first part of the event originated from areas with anomalously high SSTs (Fig. 3f; $\Delta SST_{anom} > 1°C$, relative to a 30-year climatology), suggesting that these elevated SSTs enhanced moisture uptake in the region. Further research is required to determine whether the enhanced moisture uptake resulted in an anomalously high Mediterranean contribution to precipitation totals in central Europe.

### 3.4 Forcing for vertical motion

Boris developed to the east of an upper-level PV maximum, represented here first by the PV streamer and then by its subsequent evolution as cutoff. Such regions are characterized by upper-tropospheric divergence, which evacuates air from the column and leads to compensating upward motion (Davies et al., 2024). In the case of Boris, the patterns of ascent and descent at 650 hPa are clearly discernible to the east and west, respectively, of the breaking PV streamer (Fig. 4a). QG forcing for ascent was anomalously strong for Boris, with values ranking in the top 0.16% of the all-year climatological distribution

(Fig. 4b; $\omega$ values evaluated over the red box in Fig. 1a). The high values point to a substantial contribution of the large-scale circulation to the precipitation event, differentiating Boris from other localized, "convection-driven" extreme precipitation events. The particularly favorable environmental conditions substantially contributed to the intensity and extension of the extreme precipitation event related to the storm. This might have happened in two ways: directly, by promoting ascent of moist





**Figure 2.** Large-scale flow through the evolution of Storm Boris. (left) Potential vorticity at 320 K (shaded), SLP and intersections of WCB trajectories with the 315, 320 and 325 K isentropic surfaces; (right) 6-hourly accumulated precipitation (shaded), equivalent potential temperature at 850 hPa ($\theta_e$, red dashed contours, only 300, 310 and 320 K), WCB ascent mask (orange contour; gridded WCB trajectories located between 800 and 400 hPa at the considered time), blocking mask (blue contour) and 2 PVU contour at 320 K (purple contour, smoothed using a 9-point average).





**Figure 3.** Moisture transport and sources of precipitation during the evolution of Storm Boris. (a,c,e) Integrated vapor transport (IVT) magnitude (shaded) and direction (vectors), and sea-level pressure at three different time steps; the 2 PVU contour at 320 K is shown for reference, as in Fig. 2. (b,d) Moisture sources of precipitation in the target region (red box) accumulated through the (b) 13th and (d) 14th of September 2024. (f) Mean SST anomalies for 5 to 11 Sep 2024 relative to SST climatology for September (1994-2024). Cyan contours denote the location of air parcels representing 90%, 75% and 50% of all air parcels that took up moisture over the Mediterranean and Black Sea.

air masses, and indirectly, by promoting upper-level divergence and thus intensifying the cyclonic circulation associated with

Storm Boris.




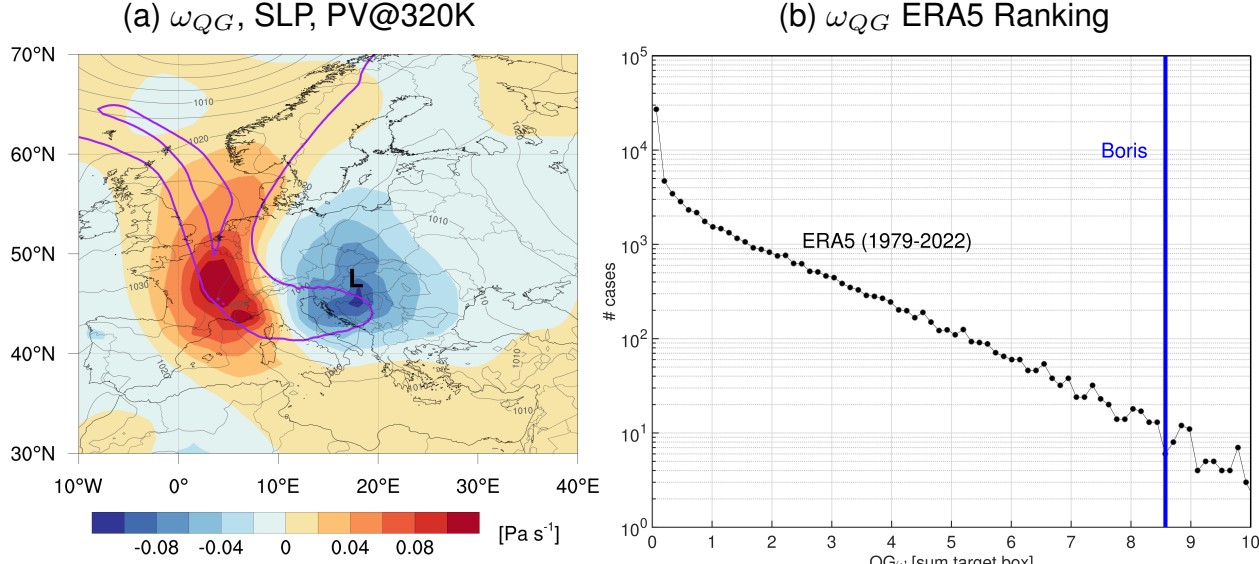

**Figure 4.** (Left) QG forcing for ascent due to upper levels for 13 September 2024 12 UTC. The QG calculation is based on operational EC analysis, and the uplift forcing is shown at 650 hPa. Only QG forcing from levels above 650 hPa are included. Positive values (red) show downward forcing, and negative values (blue) uplift forcing. The 2 PVU contour at 320 K is shown for reference, as in Fig. 2b. (Right) Comparison of maximum QG forcing (units: $10^{-2}\,\mathrm{Pa.s^{-1}}$) in target box during Storm Boris (bold vertical line, corresponding to peak in timeseries) and distribution of QG forcing during the full period 1979-2023 based on ERA5 (all-seasons confounded). Only upward forcing is considered, and the forcing is compared at 650 hPa. Note the logarithmic scale on the y axis.

### 3.5 Summary

As a conclusion of the meteorological analysis, three salient dynamical features are identified as particularly important for the unfolding of the heavy precipitation event associated with Storm Boris:

1. The formation of a slow-moving PV cutoff from the thinning and breaking of an upper-level PV streamer, being supported by a substantial contribution of upstream moist processes and by the presence of atmospheric blocking (large-scale flow evolution);

2. The track of the low-level cyclone, that allowed to concentrate moisture advection over the impacted region (low-level flow evolution);

3. The presence of anomalously strong large-scale forcing for ascent during the event (vertical motions).

This dynamical information provides the basis for the choice of the large-scale flow analogs to improve their quality, as detailed in the following section.





# 4 Dynamics-informed attribution

## 4.1 Dynamics-informed choice of the analogs

The inclusion of dynamical information in the choice of the analogs starts from the consideration that precipitation is a complex
process: it requires both moisture transport — which mostly happens at lower levels — and lifting of the moist air mass —
which is mostly driven by topography and the large-scale circulation (e.g., Doswell et al., 1996). We thus adopt a two-step
approach to constrain both the upper-level and the low-level circulation while selecting the analogs. As already mentioned in
Sect. 2.2, we use daily mean 250 hPa PV to condition the large-scale circulation over Europe and obtain a flow pattern that
resembles — in a composite sense — the one of the Boris event on the 14th of September 2024, with a PV maximum over the
central Mediterranean separated from the main PV reservoir over northern Europe (Fig. 2a). Using Z500, on the other hand,
results in a pattern that completely misses the position of the breaking PV streamer over Scandinavia, and features instead
a broader and weaker cutoff over the northern Mediterranean than the PV analogs (Fig. 5a). The "isolation" of the cutoff is
evident in the composite PV difference pattern between the two sets of analogs, in particular by the fact that the Z500 analogs
have higher PV values over central Europe than the PV analogs, while the composite derived from the PV analogs features
a separate maximum over the Mediterranean and lower PV over central Europe (Figs. 5b,d). The relative PV minimum over
central Europe also corresponds to a slightly lower precipitation in the PV analogs with respect to the Z500 analogs, resulting
from the relative anticyclonic tendency. Such a reduction in precipitation might indicate that events like Boris do not correspond
to the most extreme rainfall events that can happen over the impacted region, at least in the CESM1 model climatology: this
aspect will be further discussed in Sect. 4.3.
While the PV analogs condition the upper-tropospheric circulation to be similar to the one of the event, the lower-troposphere
may still remain relatively unconstrained, resulting in different patterns of moisture transport. Therefore, we refine the selection
of the PV analogs in each period by checking for the correct location of the surface cyclone, that has to roughly match the
one of Boris itself. Thus, an analog is selected if a cyclone center is located close to where Storm Boris was located at the
time of maximum precipitation intensity. Remarkably, from the 1050 seasonal PV analogs, only $n = 210$ refined analogs in
the period 1990–1999 and $n = 190$ refined analogs in the period 2091–2100 feature a cyclone in the same location as Boris
($n = 280$ in 1990–1999 and $n = 270$ in 2091–2100 for the yearly PV analogs). For the seasonal PV analogs, the majority of
tracks originate South of the Alps before propagating northward to the East of the Alps and enter the target region (45°N–53°N
and 16°E–25°E; see Figs. S3a,b), bearing similarity to the classic Vb track observed for Storm Boris. Similar cyclone tracks
are found in the yearly PV analogs, too, but compared to the seasonal PV analogs a larger fraction of them propagates into the
target region from the North Atlantic (Figs. S3c,d).

The so-obtained set of refined PV analogs features increased precipitation over the same region impacted by Boris, better
matching the pattern observed during the event (Figs. 5c,e; see also Fig. S4 for yearly analogs and future climate). The visual
impression is corroborated by the analysis of the euclidean distance between PV and SLP fields in the analogs and the ones
of the event, and of the Spearman spatial correlation between the analog precipitation field and the one of the event in ERA5
(regridded conservatively at the model resolution). Both quantities are computed over the same region where the analogs are





**Figure 5.** Composite mean of PV and precipitation for analogs selected using the Z500 fields, the 250hPa PV fields and the refined PV analogs. Mean 5-day cumulated precipitation (colors) and 250 hPa PV (contours) for (a) the Z500 analogs ($n = 1050$), (b) all the PV analogs ($n = 1050$) and (c) the refined PV analogs ($n = 210$). All analogs are found in the period 1990–1999 restricted to the August, September, October months (seasonal analogs). The PV contours are drawn every 0.5 PVU. Difference between (d) Z500 and all PV analogs (panel b - panel a) and (e) all PV analogs and refined PV analogs (panel c - panel b). Grid points with no significant differences (see text) are marked as white for the precipitation field and not shown for the PV field.

determined (32°-70°N, 5°W-40°E). The — not refined — set of 1050 PV analogs has a relatively small distance in the SLP field (around 0.2hPa per grid-point on average; Fig. S5) and a moderate correlation with the precipitation field of the event (around 0.35 on average). This moderate correlation arises because it is computed over an extended geographical area, and would likely be higher if its computation was restricted to the region in Central Europe with the strongest precipitation (compare also Fig. 6





and Fig. 1). The selection of the correct cyclone position in the lower troposphere improves the representation of precipitation in the analogs: the Spearman spatial correlation increases significantly from 0.35 to 0.42 ($p < 0.001$) for seasonal analogs in the present and from 0.33 to 0.36 ($p = 0.002$) in the future, and from 0.35 to 0.38 ($p < 0.001$) in the present and from 0.35 to 0.38 ($p < 0.001$) in the future for yearly analogs (Fig. S6). On average, events with a low SLP distance tends to have a better precipitation correlation, which further justifies the refinement procedure proposed here. Selected-analogs are also 4 to 8 times

more likely to have a 5-day cumulated precipitation exceeding the 75th climatological quantile and belonging to an extreme precipitation object, thus showing that the refined selection also improves the representation of the most extreme precipitation events (Fig. S7). The amplitude of the PV cutoff is also larger for the selected analogs, and the blocking activity downstream of it is enhanced, pinpointing the importance of downstream blocking for Boris' precipitation event (Fig. S8).

In conclusion, the two-step selection procedure outlined above improves the quality of the analogs of Storm Boris, both in

terms of large-scale circulation and surface impacts. In the following, unless explicitly specified, we refer to "PV analogs" as the ones obtained from the refined selection procedure. The differences in the characteristics of seasonal and yearly PV analogs between present and future model climate will be discussed in the rest of the study.

### 4.2 Future changes in Boris-like storms

The 5-day cumulated precipitation and daily mean PV fields averaged over the PV analogs for all periods are displayed in

Figure 6. The mean PV fields are close to the one of the event in all periods (cf. Fig. 2), which gives us confidence in the quality of the analogs found and the relevance of the method for representing the dynamics of Storm Boris. As shown previously with the spatial correlation, the mean precipitation fields are qualitatively similar to the precipitation field of the event, although the values are much less intense. While yearly analogs are almost similar in both their PV and precipitation fields in both periods — except for a small increase in mean precipitation over Austria — there is a large drying in the future for seasonal analogs

(Figs. 6c,f). Such a drying is accompanied by a decrease of roughly 2 PVU in the depth of the PV cutoff compared to its environment, a weakening which is also paired with a shallower surface cyclone (Fig. S9a) and reduced downstream blocking activity (Figs. S10a-c).

Because of the large number of analogs, we can additionally look at other quantiles of the conditional distribution of precipitation, especially to extremes. When considering the 90th quantile of the 5-day cumulated precipitation, the seasonal analogs

do not display the same drying as the mean which shows that the higher quantiles of the conditional precipitation distribution respond differently (Figs. S11a-c) . The yearly PV analogs, conversely, show an intensification of the response of the 90th quantile over the impacted region compared to the mean (Figs. S11d-f). This shift is reflected in the different WCB activity in the analogs, which features no significant change for seasonal analogs but an increase for yearly analogs (Fig. S12). Given the small fraction of analogs which present WCBs over the impacted region (around 10%), and the local connection between

WCBs and extreme precipitation (e.g., Catto et al., 2015; Heitmann et al., 2024), we speculate that WCBs occur during the strongest events and, hence, their change is coherent with the one of the 90th quantile of precipitation.

The different sign of the precipitation change between seasonal and yearly analogs can be understood as a change in the seasonality of Boris-like events. Seasonal PV analogs are uniformly distributed during the months of August, September,



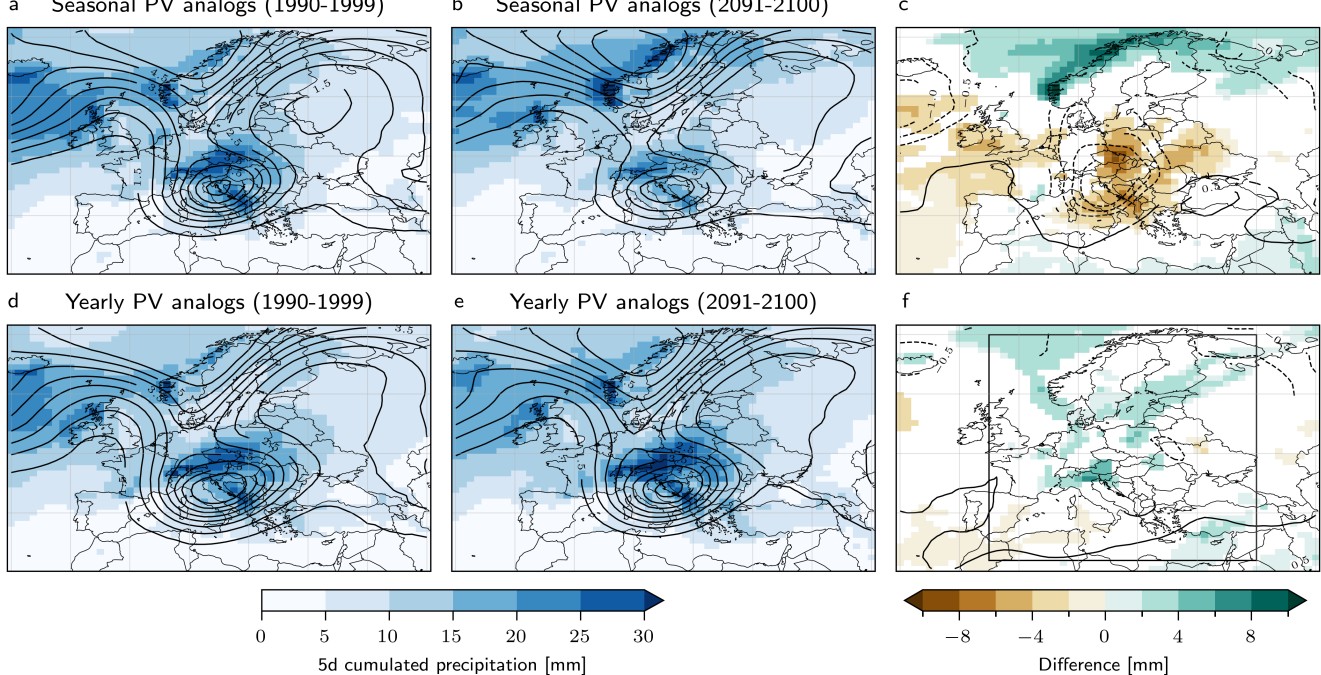

**Figure 6.** Change in the mean precipitation and PV fields of the analogs between the two periods. Mean 5-day cumulated precipitation (colors) and 250 hPa PV (contours) for the (a,b) seasonal and (d,e) yearly PV analogs in the CESM1 simulation in the periods (a,d) 1990–1999 and (b,e) 2091–2100. The PV contours are drawn every 0.5 PVU. Difference between the two periods for (c) seasonal and (f) yearly analogs. Grid points with no significant differences (see text) are marked as white for the precipitation field and not shown for the PV field. For PV, the difference is made after removing a spatial average over the domain shown to take into account the systematic shift in PV with global warming.

October months (Fig. S6c). On the contrary, there is a significant ($p < 0.001$) seasonality shift for yearly PV analogs, measured

by a $\chi^2$ contingency test to estimate changes in frequencies (Fig. S6f). While in the present 29% of analogs are found in the JJA period, only 19% are found in JJA in the future. There is also a decrease of analogs found in the SON period (from 31% to 24%) which is compensated by an increased frequency in DJF (from 8% to 14%) and MAM (from 32% to 42%). This leads to a different 2-m air temperature warming response for seasonal and yearly PV analogs, the latter warming almost half as rapidly as the former (+7.2 K vs +3.6 K, see Fig. S13). This shift has important thermodynamical consequences, which will be

discussed in Sect. 4.4.

   To investigate more systematically the evolution of cumulated precipitation between the two periods, we compute the spatial mean 5-day cumulated precipitation over the impacted region (47.2°-51°N and 12°-19.5°E) for the analogs (Figs. 7a,b). None of the analogs precipitation come close to the actual event, with a difference between the most intense analog events and the observed Boris event of 50 mm. The mean precipitation decreases for seasonal analogs (-5.6 mm, $p < 0.001$) while it does



not change significantly for yearly analogs (+2.5 mm, $p = 0.06$). There is, however, a shape change in the future towards an increase in the variance of +34% ($p = 0.2$) for seasonal and +57% ($p < 0.001$) for yearly analogs. To quantify the change in the probability of an event as intense or more than Storm Boris, we fit a Gamma distribution with the maximum likelihood method to the analogs distribution of precipitation in each period and for each analog type. The choice of the Gamma distribution was made for its simplicity of estimation and because it is commonly used to represent precipitation distributions (Martinez-

Villalobos and Neelin, 2019). To estimate the sensitivity of the estimation we estimate the probability of an event as intense as Boris over 1000 resampled analogs — with replacement — of precipitation data, always fitting a Gamma distribution. This translates into an increased probability of observing an event such as Storm Boris for both seasonal and yearly analogs by a factor 10 to 100, although the probability remains very small in both periods, typically lower than $10^{-3}$ (Figs. 7e,f). As the intensity reached by the actual event is much higher than anything observed in the model, these probabilities should be taken

with great care because they depend strongly on the parametric extrapolation with the Gamma distribution. It is nonetheless clear from the histograms of the analogs precipitation that the change in the variance leads to increased probabilities of the most intense events in the future despite the fact that the mean may not change much (yearly analogs) or even decrease (seasonal analogs).

### 4.3   Comparison with an unconditional approach

We now compare the results obtained using flow analogs with the distribution of the yearly and seasonal maximum of the spatially averaged 5-day cumulated precipitation in the model for each period, conditioning only on the occurrence of extreme precipitation regardless of the presence of a similar dynamics as the one observed during Storm Boris (Figs. 7c,d). While there is a significant increase in the mean only for yearly maximum events (+1.1mm ($p = 0.06$) for seasonal maxima and +5.4mm ($p < 0.001$) for yearly maxima), there is a similar increase in the variance of the distribution of maximum precipitation in both

periods (+70% ($p < 0.001$) for seasonal maxima and +58% ($p < 0.001$) for yearly maxima). To quantify the probability of an event as intense or more than Storm Boris we use the previously described bootstrap procedure, but this time fitting a GEV distribution as we now consider block maxima (where such a distribution can be expected, cf. Coles, 2001). This also leads to the most intense events becoming more likely: a similar event as Boris is between 100 and 1000 times more likely in the future (Figs. 7g,h). As previously, these estimated probabilities should be taken with caution because (i) it is not clear that a GEV

distribution is adapted to fit yearly maximum precipitation (Alaya et al., 2020) and (ii) none of the events in the model come close to the precipitation of Boris — which also questions the reliability of the model to represent such intense events.

Although the dynamics of unconditioned precipitation maxima has some similarities with that of Storm Boris, there are also important differences. In contrast to the Boris case, the PV cutoff is not fully isolated from the main flow and the intensity of the anticyclonic structure downstream is lower (Figs. S16a,b,d,e). The precipitation fields generally bears a large spatial similarity

with the event, with an increased intensity compared to the analogs mean as one could expect when considering seasonal and yearly maximum events. The change in the dynamics for these events is somewhat similar to the PV analogs found earlier: for seasonal maxima the precipitation field is not associated to major changes in the impacted region while the PV field sees a small reduction in its intensity in the future (Fig. S16c). For yearly maxima events, on the other hand, the PV field does not



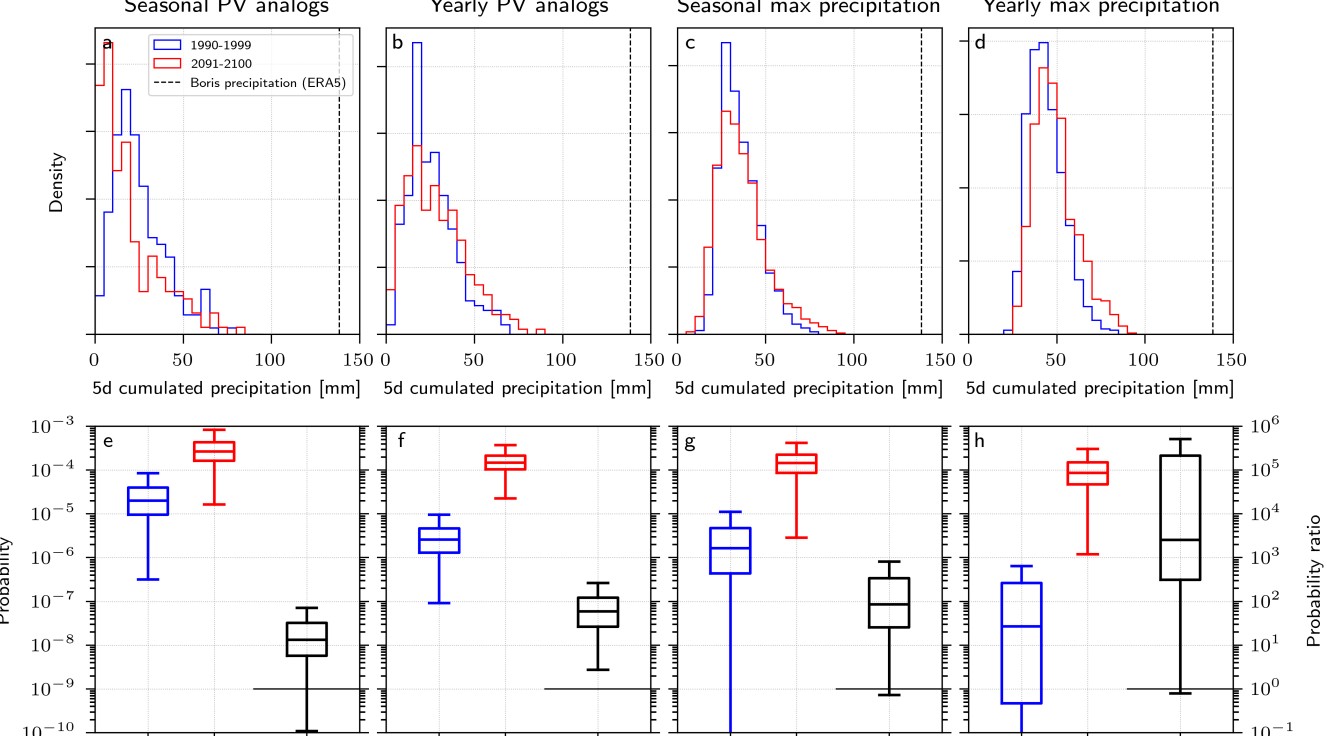

**Figure 7.** Distribution of mean precipitation in the impacted region (47.2°-51°N and 12°-19.5°E) for analogs and maximum events. Distribution of spatial mean 5-day cumulated precipitation over the impacted region for the (a) seasonal and (b) yearly PV analogs and the maximum (c) seasonal and (d) yearly events in the periods 1990–1999 (blue) and 2091–2100 (red). The number of events vary for the PV analogs (see Sect. 2.2) but not for the seasonal and yearly maximum events ($n = 1050$ in both periods). The corresponding precipitation in ERA5 for the Storm Boris over the same region are shown with a vertical dashed line. Probability to reach the ERA5 precipitation over the impacted region for the (e) seasonal and (f) yearly PV analogs and the maximum (g) seasonal and (h) yearly events in the periods 1990–1999 (blue) and 2091–2100 (red). Probabilities are estimated using a Gamma distribution for PV analogs and a GEV distribution for maximum events. The boxplots show the distribution of estimated probabilities over bootstraped samples (see text). The black boxplots show the corresponding distribution of probability ratios.

show important changes with respect to the present climate but precipitation increase strongly over the impacted region and in

the Western Balkans (Fig. S16f).

## 4.4   Extreme precipitation scaling

In the impacted region, the mean 5-day cumulated precipitation changes at a rate of -3.1%/K for seasonal and +2.7%/K for yearly PV analogs — where the mean warming is the difference between the present and the future analogs 2-m air temperature in the impacted region. For yearly PV analogs, the change is close to the mean change in precipitation in the



"dry-get-dryer, wet-get-wetter" paradigm of 2-3 %/K (Held and Soden, 2006; Byrne and O'Gorman, 2015). This is, however, not the case for seasonal PV analogs for which, despite a strong warming of +7.2K between present and future analogs, the average precipitation decreases. Even though such a decrease might look surprising, it is consistent with previous work that focused on extreme precipitation events over the Mediterranean (Armon et al., 2022) or that resulted from Vb cyclones with a track similar to Boris (Messmer et al., 2020).

We now investigate further this discrepancy between the expected increase in specific humidity under constant relative humidity with the Clausius-Clapeyron relationship of 7%/K and the actual precipitation evolution for the PV analogs using the scaling precipitation introduced in Sect. 2.4. We compute for each analog the 5-day cumulated scaling precipitation over the impacted region based on Eq. 1, and compare it with the actual precipitation in the model for seasonal and PV analogs to verify the approximate validity of the scaling (Figs. 8a,b). The precipitation values obtained from the scaling relationship are

generally close to the actual precipitation except in case of –relatively– low precipitation, which happens more often in the future period for seasonal analogs as discussed previously. This discrepancy can be expected because the extreme precipitation scaling of Eq. 1 is applied also to analogs with low or moderate precipitation, where the underlying hypotheses begin to fail (O'Gorman and Schneider, 2009). Nonetheless, the agreement with the actual precipitation in the model are quite good, and we are therefore confident in the capacity of this scaling to explain first-order changes between the present and future distributions

of precipitation. As previously, the mean and 90th quantile of the precipitation distributions are close in both periods for seasonal PV analogs but the two of them increase in the future for yearly PV analogs, the 90th quantile responding more strongly to warming than the mean for both actual and scaling precipitation (see the blue and red diamond and squares in Figs. 8a,b). For seasonal PV analogs the scaling precipitation show no decrease in the mean compared to actual precipitation in the model, another side effect of the overestimation of weak precipitation (<10 mm) by the extreme precipitation scaling.

As detailed in Sect. 2.4, we then attribute the change in scaling precipitation at each pressure level to thermodynamical changes due to the Clausius-Clapeyron relationship (change in $\Delta \frac{dq_{sat}}{dp}$), dynamical changes due to upward wind speed (change in $\Delta \omega$), change in both of those quantities at the same time and change in their covariance(Figs. 8c,d). For both types of analogs the thermodynamical response is largely positive. It is larger for seasonal analogs because of a larger warming when conditioning on the same season (+7.2K vs +3.6K at 2m) due to a seasonal shift in the yearly PV analogs (Figs. S13c,f).

On the contrary, the dynamical contribution is negative for seasonal PV analogs — meaning that the mean vertical velocity decreases between the two periods — while slightly positive for yearly PV analogs. The change in both quantities at the same time is negative for seasonal PV analogs and null for yearly PV analogs, and its amplitude is smaller than the two other terms as expected. Finally, the change in the covariance is slightly negative for both seasonal and yearly analogs, meaning that atmospheric situations with updrafts happen less often in situations with positive anomalies of $\frac{dq_{sat}}{dp}$ in the future. As a

consequence, the change in the dynamical contribution of precipitation compensate the strong thermodynamical signal in the future which leads to minimal changes in mean scaling precipitation for seasonal PV analogs (Fig. 8a). For yearly analogs — for which by definition the dynamics is closer to the one of the event — the dynamics contribution is small and the response is dominated by the thermodynamical signal, resulting in the positive increase in both scaling and actual precipitation seen in Fig. 8b.







**Figure 8.** Scaling precipitation and attribution of changes to dynamical and thermodynamical components. Mean scaling precipitation vs model precipitation for the impacted region (47.2°-51°N and 12°-19.5°E) for (a) seasonal and (b) yearly PV analogs in the periods 1990–1999 (blue) and 2091–2100 (red). The square (resp. diamond) markers show the average (resp. 90th quantile) in each period. The black dashed line shows the $y = x$ line. Scaling precipitation change between the two periods for the (c) seasonal and (d) yearly PV analogs.

From a dynamical point of view, the reduction in vertical velocity in future seasonal analogs is likely related to the upward displacement of the tropopause expected with tropospheric warming (Meng et al., 2021). This process would result in weaker horizontal PV gradients at 250 hPa (Fig. 6c) and, consequently, in a weaker differential vorticity advection at upper levels (a



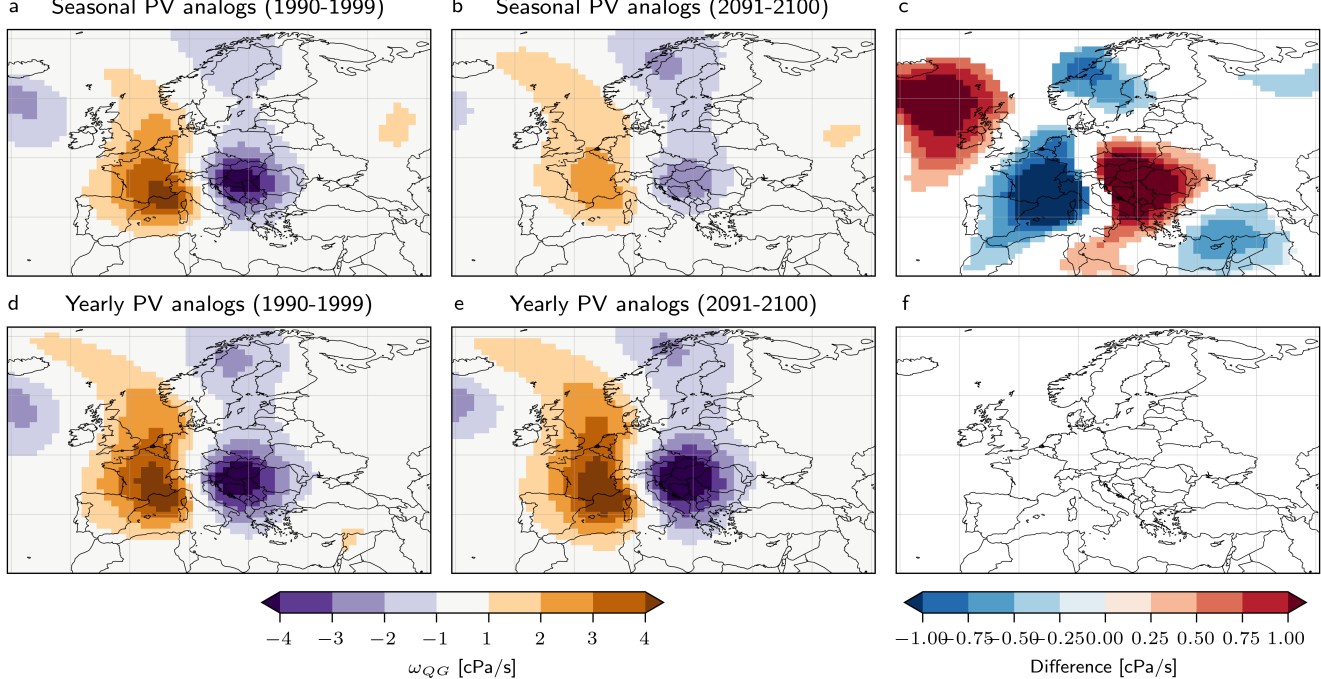

**Figure 9.** Change in the mean QG vertical wind speed. Mean QG ascent at 650 hPa due to upper levels for the (a,b) seasonal and (d,e) yearly PV analogs in the CESM1 simulation in the periods (a,d) 1990–1999 and (b,e) 2091–2100. Only QG forcing from levels between 650 hPa and 100 hPa is shown. Positive values show downward forcing and negative values uplift forcing. Difference between the two periods for (c) seasonal and (f) yearly analogs. Grid points with no significant differences (see text) are marked as white.

process forcing vertical motions in the QG-omega equation, see, e.g., Davies, 2015). To check this mechanism we compute the vertical wind speed of analogs predicted by the QG-omega equation in the mid-troposphere (650 hPa), and average it over the analogs to estimate the large-scale forcing for ascent resulting from the upper-level flow. As expected from the change in the PV fields between the two periods, the forcing due to upper levels is much weaker in the impacted region in the future for seasonal analogs, while it does not significantly change for yearly analogs (Fig. 9). The forcing due to lower levels, on the other hand, shows minimal change for both seasonal and yearly analogs, further highlighting the role of upper-level circulation changes (Fig. S14). We additionally computed the scaling equation using the QG-omega vertical wind (Fig. S15) instead of the actual wind and found similar results as in Fig. 8, with the important difference that scaling precipitation using the QG omega wind tends to be 2-3 times smaller because of the underestimation of the vertical wind velocity by the QG approximation.

Finally, we notice that a similar difference in scaling between seasonal and yearly events also exists for the unconditioned extreme precipitation events, which feature a rate of change of precipitation with respect to warming of only +0.7%/K for the seasonal and of +6.8%/K for the yearly maxima of precipitation (not shown). While the change in yearly maximum events is close to the expected Clausius-Clapeyron scaling of +7%/K for the strongest events, restricting to the seasonal maximum





show a much lower change in maximum precipitation. This likely reflects a change in dynamics similar to the one observed for seasonal and yearly PV analogs, with a lower intensity of the PV cutoff in the future for seasonal maximum events (as can be inferred from Fig. S16).

In conclusion, the outlined results support our interpretation that dynamical changes in the large scale forcing for ascent are
able to substantially modulate the occurrence of extreme precipitation related to upper-level PV cutoffs, as in the case of Storm Boris.

## 5   Discussion

In this piece of work, we have framed the question of the effect of anthropogenic global warming on Boris-like storms in the context of different attribution approaches, employing both so-called 'conditional/dynamical' and 'unconditional/probabilistic'
methods. In this section, we would like to draw more general insights about the differences between these methodologies, with the aim to inform future research work relying on these approaches.

Although the attribution question — has the likelihood/intensity of a Boris-like storm increased under climate change? — seems simple at first glance, the way such a question is framed in practice can influence the answer to a large extent. Aside from model uncertainties and data limitations, the ambiguity of the question lies first of all in what one means by 'Boris-
like storms', as already pointed out by several studies (Cattiaux and Ribes, 2018; Naveau et al., 2020), which showed that how the event is defined spatially and temporally greatly influences the results of the attribution. This is rather intuitive when employing the unconditional approach (based on, e.g., GEV fits): for instance, a too large or too small spatial area to define the event could blur the severity of the event with respect to the background. Here we investigate another ambiguity related to the definition of the event: namely whether we should compare events with similar intensities or similar dynamics. To do
so we applied both a conditional and an unconditional approach to the same climate model data. If one defines 'Boris-like storms' by conditioning on events with the same atmospheric dynamics and occurring in the same season as the event, then we showed that on average they are expected to lead to *less* intense precipitation in a warmer world — although the precipitation of the analogs with the most intense precipitation (90th quantile) do not change significantly. If such 'Boris-like storms' are allowed to be found in other seasons, however, then they are more intense both on average and for the most extremes — the
increase in the extremes being larger than for the average. If one frames the question as in probabilistic attribution methods — i.e. without any conditioning on the actual dynamics of the event and focusing only on the intensity of precipitation — and therefore understand 'Boris-like storms' as events producing at least similarly intense precipitation, then this kind of events are expected to become more intense if they occur in the same season, and even more intense if they occur in any season.

Conditional and unconditional attribution methods answer different questions and serve different purposes. The uncondi-
tional approach is likely the most useful with regards to impacts for the layperson, who may not be interested in whether the next impactful storm will be similar to Boris but whether they can expect more or less intense precipitation from it. However, one does not necessarily need to observe a storm like Boris to answer this question: in this regard, the probabilistic approach only uses the observation of an event as a pretext, without addressing the peculiarities of the actual observed event. This may



lead to a form of selection bias when evaluating the evolution of extreme events with climate change (Barlow et al., 2020; van Oldenborgh et al., 2021; Miralles and Davison, 2023). In contrast, conditional attribution methods do consider the dynamics of the event observed and try to assess how an event with similar dynamics would unfold in a warmer/colder world. In this sense they may be less relevant for the layperson, but be more understandable and scientifically accurate than the unconditional approaches, which pose the risk of implicitly comparing events with very different dynamics. The example of Storm Boris shows it clearly, because Boris-like events do not necessarily overlap with the events of maximum precipitation in the model climatology: consequently, any attribution statement based on the latter set of events will actually not be relative to events "like Boris". This mismatch may muddle our understanding of how global warming has affected or will affect this type of extreme event: for instance, how could one reliably connect potential trends in the most extreme precipitation events over central Europe to trends in the occurrence of PV cutoffs over the Mediterranean, if such extreme events are not actually related to PV cutoffs? Thus, we believe that attribution studies of real-world events would profit of a more systematic comparison between methods and of an explicit consideration of event dynamics, like the approach employed in this study.

Our results might also have some implications for other conditional attribution methods, such as nudging or PGW. Those two methods can impose the same dynamics as the observed event in a future/past climate, and simulate its evolution with a weather or climate model. The use of analogs, on the other hand, selects events that are as close as possible to the one observed in terms of dynamics, without imposing any external constraint on the model itself. If such events with similar dynamics show a systematic circulation shift between the two periods — i.e., if the closest analogs of the event are not as close in one period compared to the other — this allows to directly recover dynamical changes. As discussed previously, we showed that there are substantial changes in the dynamics of Boris-like storms for analogous events occurring in the same season, with less intense PV cutoffs leading to less intense large-scale forcing for ascent — measured here with the QG vertical wind — which in turns leads to a dynamical damping of extreme precipitation. We believe that representing such a large change in dynamics (i.e., a drop of the central PV in the cutoff from ∼5 to ∼3 PVU, see Figs. 6a,b) might be challenging for nudging-like methods. Consequently it becomes possible that, by forcing a similar dynamics in both the factual and counterfactual periods, these methods compare events with vastly different return times: in other words, a similarly deep cutoff as the one associated with Storm Boris would be way more unlikely to occur in September in the future compared to the present. The same issue might apply to more refined PGW approaches, such as using nested high-resolution model runs to performed quasi-real-time case study attribution: it is important to be mindful of this potential problem when designing such studies, and address it by an explicit treatment of dynamical changes (see, e.g., Brogli et al., 2023).

## 6   Conclusions

In this study we have employed an analog-based approach to study the unfolding of Boris-like events in a warmer climate. By relying on hundreds of simulated years from reruns of the CESM1 large ensemble with detailed model level output, we are able to circumvent the problem of lack of good analogs for extreme events that usually affects analog-based attribution studies. Furthermore, the high spatial and temporal resolution of the three-dimensional CESM1 output allowed the deployment



of sophisticated dynamical diagnostics, that gave important insights into the dynamics of Storm Boris in present and future climates: noteworthy in this regard are the correct depiction of the PV cutoff and of the Vb cyclone track related to Storm Boris, the relevance of atmospheric blocking and WCBs for the occurrence of extreme precipitation, and the quantification of changes in dynamically-driven forcing for ascent. The proposed dynamics-informed approach can be applied to other cutoff related, extreme precipitation events (which are notoriously difficult to study using analog-based approaches, as noticed by Thompson et al., 2024), with the aim to increase the robustness and the physical consistency of attribution statements and of other analog-based studies. From a physical point of view, it is noteworthy that not only the thermodynamical response due to the Clausius-Clapeyron scaling of saturation specific humidity is important for the evolution of the intensity of the event, but also the dynamical response due to changes in the properties of the upper-level cutoff low associated with the storm. We showed that this dynamical response can be strong enough to compensate the thermodynamical response and even reverse the sign of the expected positive intensity change. The understanding of this effect is complemented by the seasonality change displayed by Boris-like events in the CESM simulations, whose occurrence shifts towards the shoulder seasons — with potential implications for adaptation, because the occurrence of the same precipitation event in a different season might not necessarily lead to the same impacts (e.g., Tarasova et al., 2023).

In addition to the attribution statement for Boris-like events, we propose here three more general "lessons learnt" from the complementary dynamically-informed investigations performed in this study.

*It is possible to improve the selection of large-scale flow analogs for extreme precipitation events by including specific dynamical information*, e.g., by using upper-level PV or by explicitly checking for the correct position of the surface cyclone in the analogs. The inclusion of dynamics is particularly important for large-scale, heavy precipitation events, which cannot happen if the adequate dynamical forcing for ascent is not present. The synoptic-scale circulation also contributes indirectly to the extremeness of the event by inducing a slow-moving flow pattern (featuring, e.g., the presence of Rossby wave breaking and atmospheric blocking), that exacerbates precipitation over the same regions. Slightly different processes might be at play for mesoscale or local-scale extreme precipitation events that are most often related to trailing convection: for instance, the vicinity to the sea and the enhancement of convection by latent heat release might locally compensate the reduction in QG forcing (Llasat et al., 2007). However, the occurrence of such events still cannot completely prescind from a favorable large-scale setup: an example is the 2024 Valencia flash-flood, that occurred in correspondence to a slow-moving, upper-level PV cutoff, i.e., in a situation with dynamical similarities to Storm Boris. This consideration leads us to speculate that the representation of this type of mesoscale extreme events in analogs could also be improved by an appropriate consideration of atmospheric dynamics.

*The methodology chosen to define the extreme event of interest can substantially affect the conclusions of projection/attribution studies, and even result in opposite climate change signals*. The case of Storm Boris is emblematic, because the choice of computing yearly or seasonal analogs results in different attribution outcomes. Thanks to the comparison of different methods, we were able to reconcile the diverging results: the weakening observed when focusing on seasonal analogs can be contextualized as part of a shift in event seasonality, that emerges yearly analogs are considered. Defining a Boris-like event by relying only on the precipitation totals is also problematic, because this choice might lead to events driven by different dynamics to be implic-



itly compared, hindering the comprehension of the physical processes at play. It is remarkable, although not surprising *per se*, to notice that even in an "ideal" situation, when enough data are available to correctly identify analogs, one still cannot escape the ambiguity related to the practical definition of the event. This important limitation contextualizes the outcome of every attribution assessment with respect to the chosen methodology, with no approach being a priori better or worse than the other. That's why we believe that this study could be regarded as a sort of "meta-attribution", where methodologies are compared and their differences understood thanks to the use of dynamical diagnostics, rather than as a final attribution/projection statement about Storm Boris.

*The use of model nudging frameworks to replicate extreme precipitation events in a warmer climate should be evaluated with particular care to correctly consider the change of dynamics* — and of their coupling with the thermodynamics — between present and future climate. This work shows that, at least in the "model world" of CESM1, such changes can be substantial and may result in an over- or underestimation of the precipitation when not adequately taken into account. Simulating a Boris-like event in a future climate while nudging the same PV structure of the present-climate, for instance, would likely result in an unrealistically strong precipitation event, with return times much larger than the ones of Boris in the current climate.

*Author contributions.* EA oversaw the analysis of precipitation data, IT performed and interpreted the moisture source diagnostic, VS performed the analysis of atmospheric blocking, HB the analysis of warm conveyor belts, MS the analysis of QG forcing for ascent, KH the computation and analysis the extreme precipitation objects, and MF (together with MS) performed the cyclone identification and tracking, both in observations and model data. RN performed the analogs computation, the attribution, scaling and statistical analysis. JR coordinated the project and, together with RN, wrote the manuscript. All authors contributed to the interpretation of the results and the editing of the manuscript.

*Competing interests.* The authors declare that they have no competing interests.

*Acknowledgements.* This work stem from a collaborative effort from several colleagues in the group of Atmospheric Dynamics and was initiated by Heini Wernli, whom we thank for the initial idea and for the ongoing fruitful discussions and feedback. We thank MeteoSwiss and ECMWF for granting access to the IFS model analysis and the ERA5 reanalyses, Urs Beyerle (ETH Zurich) for performing the CESM-LE reruns as a contribution to the ERC project INTEXseas, and Franziska Aemisegger for providing the msd-cpp code for the calculation of the moisture sources. JR received funding from the Swiss National Science Foundation (SNSF) via grant no. 209135. The contribution of MF is funded by the SNSF via grant no. 196978.



**Data availability statement**

The ERA5 reanalyses (Hersbach et al., 2020) can be obtained from the Copernicus Climate Change Service, Climate Data Store
(https://doi.org/10.24381/cds.adbb2d47). The code for the CESM model (Hurrell et al., 2013) that was used in the CESM-
LE simulations is available from https://www.cesm.ucar.edu/models/cesm1.0/ (last access: 20 September 2024). The model
outputs of the CESM-LE reruns, together with the climatologies of blocking, warm conveyor belts, extratropical cyclone and
QG-forcing for ascent used in this study (both in the model and reanalysis) are available from the authors upon request.



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
