# Peer review of "Storm Boris (2024) in the current and future climate: a dynamics-centered contextualization, and some lessons learnt"

_EGUsphere, 2025_

## Referee Comment (RC1)

This study performs a process-understanding projection of Storm Boris, with a very interesting approach that combines climate and weather information. The paper also shows the potential of a methodology to disentangle the dynamic and thermodynamic signals of climate change on specific events and assess their contributions separately. This stands in contrast to other methodologies that either ignore the dynamical information of the event or constrain it so tightly that part of the signal is lost. I very much enjoyed reading the paper; the structure and language are very clear and easy to follow.

A concern relates to the model performance in simulating mesoscale extreme events within this type of cyclones. Other studies using the same model (Dolores-Tesillos et al. 2023; Karwat et al. 2024; and from some of the authors themselves Binder et al. 2024; Joos et al. 2024) have raised questions in this regard. I understand that, as the authors state, this work can be seen as a "meta-*attribution*"\* study rather than a definitive *attribution* analysis. However, I think it is important to discuss whether CESM is suitable for simulating this type of storm. My understanding is that the model does not reproduce weak cyclones very well, especially in the Mediterranean. Could this limitation influence your results? Also, more generally, what are the limitations of using a single model, and how do the authors justify using a high-end, unrealistic, rcp scenario (Hausfather & Peters 2020)?

\*I understand that the authors apply a methodology within the framework of extreme event attribution and therefore repeatedly frame the study in terms of "*attribution*." However, in my view this may be somewhat misleading for communication purposes, and in several places, I would suggest rephrasing in terms of "projection." In the attribution literature, "counterfactual" typically refers to a climate that could have occurred (without anthropogenic forcing) or that might occur under d2ifferent forcing conditions, while "factual" refers to the presetn climate. For projections, the concept of "future counterfactuals" is also used (e.g., Ermis et al. 2024). It is therefore confusing to use "counterfactual" for the present climate and "factual" for the future climate (e.g., lines 136–138). A clarification of terminology here would be very helpful.

Other points that would benefit from justification or clarification:

- Choice of the domain. It seems quite large; why do you need to include the main PV reservoir over Scandinavia? As you state in the discussion, the domain can strongly influence results. Have you performed sensitivity tests, e.g. focusing only on the cut-off low?
- I understand (and like) the use of PV as the first step to define the event. However, as the authors state, most analog studies use slp or geopotential height, and storm Boris had a peculiar track of the surface cyclone. Have you considered, for comparison, first identifying analogues of SLP (or Vb tracks, as in Ginesta et al. 2024, using the same dataset) and then constraining them with a cut-off low based on PV? Do you think this would yield more/better (or fewer/worse) analogues, or perhaps a stronger precipitation correlation?
- Am I correct in assuming that you compare with Z500 analogues here because the SLP signal of this cyclone is too weak for a meaningful comparison?
- Can you justify the choice of the domain of target/analogue slp identification (red box)?
- Line 328: should analogues also be defined at the time of maximum precipitation intensity, or is this criterion applied only for Boris itself?
- In the discussion/conclusion you argue that simulating Boris in a future climate while nudging the present-climate PV structure would yield an unrealistically strong precipitation event, so nudging frameworks should be applied with care. I agree. However, isn't that precisely the purpose of some storyline approaches—to strongly constrain dynamics in order to isolate the thermodynamic response? I see your point that a full picture of climate-change effects requires considering both dynamics and thermodynamics, but I think it would be worth clarifying how you distinguish your approach from these alternative frameworks.

Typos:

Line 315: Fig 2a. I think is Fig. 2b

Line 322: reduction → difference?

Line 328: is 'close' referring to the target region?

Line 333: Fig S3 I cannot identify where storms are born; is it possible to highlight cyclogenesis?

Line 580: 'that emerges yearly analogs' → emerges *when* yearly

Links to references

Hausfather & Glen P. Peters, 2020 https://www.nature.com/articles/d41586-020-00177-3

Dolores-Tesillos et al. 2022 https://wcd.copernicus.org/articles/3/429/2022/

Karwat et al. 2024 https://journals.ametsoc.org/view/journals/clim/37/4/JCLI-D-23-0160.1.xml

Ermis et al. 2024 https://iopscience.iop.org/article/10.1088/2752-5295/ad4200

Ginesta et al. 2024 https://journals.ametsoc.org/view/journals/clim/37/21/JCLI-D-23-0761.1.xml

---

## Referee Comment (RC2)

**Review of "Storm Boris (2024) in the current and future climate: a dynamics-centered contextualization, and some lessons learnt"**

By Jacopo Riboldi, Robin Noyelle, Ellina Agayar, Hanin Binder, Marc Federer, Katharina Hartmuth, Michael Sprenger, Iris Thurnherr, and Selvakumar Vishnupriya

Submitted to: Weather and Climate Dynamics (egusphere-2025-3599)

The study examines Storm Boris, which caused heavy rainfall and severe flooding in central Europe. In particular, it attempts to understand how such storms may change as a result of climate change. Various methods are used, including a conditional (dynamic) method and a probabilistic method. The paper thus raises an important scientific question: to what extent does the method used to attribute climate change alter the result? It becomes clear that different processes (thermodynamic and dynamic) can have different and even opposite effects on such complex systems, as is the case for Storm Boris. The comprehensive analysis is a novelty and emphasizes that it is important to know the exact research question, but also that a comprehensive analysis is important in the attribution of climate change. The paper is well-structured and easy to follow. Therefore, I would suggest accepting this study after minor revisions.

Minor comments:

A general comment regarding CESM. I do not understand why the authors assume that CESM would be able to generate the same precipitation intensity as the clearly higher resolved (spatially and vertically) ERA5 data. To be fair, you would need to downscale the selected analogues to a resolution that resembles the one of ERA5. Therefore, I suggest excluding the paragraph around L390 to 403 and the second row of Fig. 7 (e-h).

L11: What exactly are 'Boris-like' events?

L16: ad -> at

L17: "–," is a bit of a strange notation

L18ff: Starting at: The results obtained … You do not specify what the results are of the unconditional method, which I think is interesting and important. I understand that the abstract is already quite long, but I think it is an important part of the study. Therefore, I suggest writing the first part of the abstract a bit more concisely, so that there is room for these results too.

L37f: Changing dynamics could also mean that the tracks of such Boris-like storms are less or more frequently used.

L40f: This sentence is rather long. I would recommend splitting and shortening it.

L58f: The description of the path sounds more like a 'Vc' rather than a 'Vb' track.

L102: European Centre for Medium Range Weather Forecasts -> ECMWF

L111: I am wondering if the 30 vertical levels might not be a limiting factor for your analysis. I will point out some locations later in the results again.

L126: The first figure you are mentioning is Fig. 6. Generally, it is expected that the numbering starts with the first figure.

L129: You are using PV at 250 hPa. What interpolation method from model to pressure levels was used for CESM data?

L130: What is the rationale behind choosing September 14th to represent the dynamics of Storm Boris when identifying analogues? Can you give an indication on how sensitive your results are with respect to this selection?

L132: "Furthermore, for each daily mean PV field, we subtract the spatial mean of the PV field …" This sentence is unclear to me. Do you subtract the daily spatial mean or some climatology?

L150: The bootstrapping method is unclear to me. Do you sample a 1000 times 1050 analogues? Maybe you can rephrase this part a bit to increase understandability.

L166: You mention that you are lowering the PV anomaly threshold, but it is the same number (-0.7 PVU) as in line 163. This is unclear.

L159: How can you obtain a resolution of 20 hPa in CESM with only 30 vertical levels?

L189: 1.5 x 1.5 -> 1.5° x 1.5°

L195: I would not consider the 75th percentile as extreme precipitation. It might be strong precipitation, especially when considering the long simulation runs of CESM.

Section dynamical diagnostics: It is unclear why you are applying some of the tools/algorithms to operational ECMWF data and to ERA5 reanalysis.

L231: The fact that the impact region is more over eastern Europe could imply that Boris was more like a Vc cyclone.

L234: From what source/dataset do these 350 mm come from?

L246: Can you identify if the uplift of the 600 hPa is triggered by dynamics or orography?

L281: If I understand Fig. 3f correctly, all of the Mediterranean has an SST anomaly of more than 1 °C.

L289: What is meant by "all-year climatological distribution"? Is this based on daily means?

L328: "… is selected if a cyclone center is located close to where Storm Boris …". How is close defined?

L332:  South -> south

L335: Does "propagate into the target region from the North Atlantic" mean that the track passes by the Alps only on the northern side only?

L338ff: I do not understand this sentence.

L349: Also, the fact that Vb cyclones are only characterised by the track of their surface position justifies this refinement.

L369f: "the seasonal analogs do not display the same drying as the mean" This sentence is not very clear, as it could indicate that the drying pattern just looks different, but in fact, there is almost no drying visible.

L374ff: Could the small fraction of WCB detected also be related to the comparably low vertical resolution of CESM with only 30 levels?

L383f: This sentence is difficult to understand. First, you mention seasonal and yearly analogues, then you refer to the warming of the yearly and then the seasonal, but the numbers in the brackets are again for seasonal and then yearly. I suggest reorganising this sentence to make it more intuitive.

L386: I suggest using the word accumulated precipitation instead of cumulated precipitation. You will have to replace this on multiple occasions.

L388f: Why do you expect the same precipitation intensity between ERA5 and CESM? The resolution of these two data sets is quite substantially different, making it very hard for CESM to actually reach the same intensity as in ERA5. This is even true if you would regrid ERA5 to the CESM grid, as the underlying resolution is still much higher.

L389ff: "The mean precipitation decreases" with respect to what?

L391 to the end of the section: It is unclear to me why you are assuming that CESM should be able to capture the same intensity as ERA5. Therefore, I do not see the point of this analysis here.

L395: There is 'estimate', 'estimation' and 'estimate' in one sentence. Please rephrase

L415: Again, I think part of this is due to the resolution of CESM. It would be interesting to know how precipitation intensity increases for these events if downscaled to the same resolution as ERA5, which, of course, is beyond the scope of this study.

L445: Why are you looking at the 90th percentile here and not at the 75th percentile as mentioned in the method section?

L519ff: This sentence is unclear to me.

L545: I agree that CESM has a high temporal resolution with 6-hourly output, but the spatial and especially the vertical resolution is not really high.

L580: "…, that emerges yearly analogs are considered." This part does not seem to fit into this sentence.

Figure 2: The difference between the blue and purple contours is difficult to see, at least in a printed version. Why is PV shown at 320 K and not at 250 hPa, as it was introduced in the method section?

Supplementary Figures: Make sure that they appear in an increasing order in the text. I think S16 is mentioned before S15 and S14.

---

## Author Comment (AC1)

We would like to thank the two reviewers for the constructive feedback, because addressing their comments allowed us to improve both scientific consistency and presentation clarity. Reviewer comments are written in black, while replies are marked in bold green.

**Reviewer 1**

This study performs a process-understanding projection of Storm Boris, with a very interesting approach that combines climate and weather information. The paper also shows the potential of a methodology to disentangle the dynamic and thermodynamic signals of climate change on specific events and assess their contributions separately. This stands in contrast to other methodologies that either ignore the dynamical information of the event or constrain it so tightly that part of the signal is lost. I very much enjoyed reading the paper; the structure and language are very clear and easy to follow.

We thank the reviewer for their positive evaluation of our work and address their comments below.

A concern relates to the model performance in simulating mesoscale extreme events within this type of cyclones. Other studies using the same model (Dolores-Tesillos et al. 2023; Karwat et al. 2024; and from some of the authors themselves Binder et al. 2024; Joos et al. 2024) have raised questions in this regard. I understand that, as the authors state, this work can be seen as a "meta-attribution"\* study rather than a definitive attribution analysis. However, I think it is important to discuss whether CESM is suitable for simulating this type of storm. My understanding is that the model does not reproduce weak cyclones very well, especially in the Mediterranean. Could this limitation influence your results?

It is true that, as noted by the reviewer and given its coarse resolution, we do not expect that the CESM1 model used here reproduces correctly the climatology of mesoscale extreme precipitation events. However, we have shown that for the Boris storm a large part of the forcing for ascent comes from large-scale, quasi-geostrophic (QG) forcing. This mechanism is resolved at the resolution of the model and therefore we can reasonably expect the model to represent correctly how this would change in a warmer climate. Moreover, even if the model cannot reproduce the climatology of weak cyclones in the Mediterranean or cut-off lows in Central Europe, we expect this bias to cancel out because we compare two periods inside the same model and, therefore, to obtain a reasonable estimate of the change between the two periods. This is especially true when conditioning on similar atmospheric patterns as the one observed during the Boris storm: while the model may be biased in how it represents the frequency of cut-offs and Vb cyclones, we may also expect that its physics is reasonably close to the truth conditional on the

observation of a cut-off and a Vb cyclone. We propose to add a new subsection "Limitations" to Sect. 5 and explicitly discuss these limitations with the following:

"The CESM1 climate model used here to evaluate changes in Boris-like storms has a relatively coarse resolution and its capacity to simulate mesoscale extreme events can therefore be questioned, as already pointed out by previous works (Dolores-Tesillos et al. 2022, Joos et al. 2023, Karwat et al 2024, Binder and Wernli 2025). We do not expect the model to reproduce correctly the climatology of mesoscale extreme precipitation events, nor the intensity of extreme precipitation (see our discussion of Fig. 7). However, we have shown that, for the Boris storm, a large part of the forcing for ascent comes from the large-scale forcing of the PV cutoff. This mechanism is resolved at the resolution of the model and therefore we can reasonably expect that its evolution in a future climate with weaker cutoffs is well reproduced. Moreover, even if the model cannot reproduce the climatology of weak cyclones in the Mediterranean or cut-off lows in Central Europe, because we compare two periods inside the same model, we expect the bias with respect to the real world climatology to cancel out to give a reasonable estimate of the *change* between the two periods. This is especially true when conditioning on similar atmospheric patterns: the model may be biased in how it represents the frequency of cut-offs and Vb cyclones but we may also expect that its physics is reasonably close to the truth conditional on the observation of a cut-off and a Vb cyclone."

Also, more generally, what are the limitations of using a single model, and how do the authors justify using a high-end, unrealistic, rcp scenario (Hausfather & Peters 2020)?

We agree that the RCP5-8.5 scenario is a high-end, economically unrealistic warming scenario. However, by imposing a high-warming scenario to the climate system, we also expect to improve the signal to noise ratio when detecting dynamical changes in weather systems, which are expected to be much smaller than the thermodynamical changes. Using a single-model is indeed also a major limitation – especially when it comes to investigating highly-parametrized precipitation – but few models have available 1000 years of 3D data at 6 hourly resolution as the one we are using here for our analogs and precipitation scaling analysis. These points will also be discussed in the new subsection "Limitations" in Sect. 5.

I understand that the authors apply a methodology within the framework of extreme event attribution and therefore repeatedly frame the study in terms of "attribution." However, in my view this may be somewhat misleading for communication purposes, and in several places, I would suggest rephrasing in terms of "projection." In the

attribution literature, "counterfactual" typically refers to a climate that could have occurred (without anthropogenic forcing) or that might occur under different forcing conditions, while "factual" refers to the present climate. For projections, the concept of "future counterfactuals" is also used (e.g., Ermis et al. 2024). It is therefore confusing to use "counterfactual" for the present climate and "factual" for the future climate (e.g., lines 136–138). A clarification of terminology here would be very helpful.

We agree with the reviewer that this may be confusing, despite the fact that the separation between factual and counterfactual periods is *a priori* arbitrary. As suggested, we will use through the manuscript the term 'future counterfactual' for the simulations with the future RCP.

Other points that would benefit from justification or clarification:

- Choice of the domain. It seems quite large; why do you need to include the main PV reservoir over Scandinavia? As you state in the discussion, the domain can strongly influence results. Have you performed sensitivity tests, e.g. focusing only on the cut-off low?

We agree that the domain chosen to find analogs is large, including the PV reservoir over Scandinavia. We chose this domain because, as argued in section 3, the Boris event is characterized by the detachment of a PV cut-off situated between two large anticyclones, whose growth and motion was partly responsible for the thinning and the successive breaking of a PV streamer. The choice of this large domain is consequently influenced by the need of appropriately resolving such high-latitude anticyclones. We emphasize, however, that the analog refinement procedure based on the position of the low-level cyclone strongly filters the analogues found, so that changing marginally the size of this large domain has few consequences on the results. We also tried to use a much smaller domain centered on the cut-off and found similar results for precipitation (see Figure R1.1 below), especially the reduced intensity of the cutoff, but the analogs featured a strongly anticyclonically tilted PV streamer and did not feature the broad anticyclone over western Russia and, thus, did not adequately represent the large-scale dynamics during the event.

Figure R1.1: in the first two columns, as in Figure 5 of the manuscript, are shown precipitation and 250hPa PV composites for analogs computed over a smaller region centered on the cut-off location (red box in sub-plots a,b). The first row depicts analogs in present climate, second row in future climate and third row their difference. In the third and fourth column, for comparison, analogs are computed over the same large region used in the study. Here we show all 1050 analogs with no refinement.

- I understand (and like) the use of PV as the first step to define the event. However, as the authors state, most analog studies use slp or geopotential height, and storm Boris had a peculiar track of the surface cyclone. Have you considered, for comparison, first identifying analogues of SLP (or Vb tracks, as in Ginesta et al. 2024, using the same dataset) and then constraining them with a cut-off low based on PV? Do you think this would yield more/better (or fewer/worse) analogues, or perhaps a stronger precipitation correlation?

The methodology proposed by the reviewer would essentially be the reverse of the one we employed: selecting analogues of low-level cyclones first and then filtering those associated with mid-level PV cut-offs. This is indeed something that is feasible in general, and we do expect that it would give similar results as the one used here (because the intersection of two sets is independent of which set is considered first). However, for this particular case we think it may present some difficulties, mainly because we are working in a region with

strong topography where the use of the SLP to define low-level cyclones could be problematic (see our discussion in Sect. 2.3 about the tracking of the low-level cyclone for Boris) and lead to spurious analogues. In Ginesta et al. 2024 the low-level cyclone occurred over the sea or close to the coast, where the use of SLP is better justified. We will discuss the alternative procedure and the potential issues related to orography in the revised version, proposing this formulation:

"Note that the reverse procedure could also be implemented, first selecting on SLP analogs and in a second step refining the selection with events featuring an upper-level PV cutoff. We expect that the two procedures would give similar results: however, given that the complex orography of the impacted region might lead to spurious SLP patterns, basing the initial selection on upper tropospheric features seems a preferable option."

- Am I correct in assuming that you compare with Z500 analogues here because the SLP signal of this cyclone is too weak for a meaningful comparison?

Yes, this is indeed one of the arguments. Another reason is that the cut-off signal would not be seen on SLP whereas it is clearly seen on Z500 (and even more visible in the PV field), and, as discussed above, we start by finding analogues of the cut-off low.

- Can you justify the choice of the domain of target/analogue slp identification (red box)?

The target region for identifying the low-level cyclone is not a domain where we compute SLP analogues, but rather where we impose the center of the low-level cyclone to be located. The size and location of the box are based on the regions used by Messmer et al. (2015) in their Vb cyclone identification to constrain the end locations of Vb cyclones (46N–52N, 12E–19E). We shifted the location of their original box slightly northeast (45N–53N, 16E–25E) to better capture the track of storm Boris, which propagated unusually eastward compared to a textbook Vb cyclone. To simplify the analysis and interpretation with respect to Messmer et al. (2015), we also dropped the requirement that the cyclone tracks must end in the box, and instead just require that a cyclone center must be present in the box at the time step of the analogue.

We propose to add the following clarification:

"Therefore, we refine the selection of the PV analogs in each period by checking for the presence of a surface cyclone in the region where Boris propagated through. This target region is based on the Vb cyclone track end constraint used by Messmer at al. 2015, and shifted northeastward to account for the track of storm Boris (45N–53N, 16E–25E; see Fig. S3)".

- Line 328: should analogues also be defined at the time of maximum precipitation intensity, or is this criterion applied only for Boris itself?

Thank you for pointing that out, we realize that this sentence might be misleading. Indeed, the box to filter low-level cyclone is defined with respect to where the center of the low-level cyclone was at the day of maximum precipitation during the Boris storm, but we do not impose that analogues need to have the center of their low-level cyclone in this box AND the day of their maximum precipitation at the same moment. We precise it as follows: "Thus, an analog is selected if a cyclone center is located close to where Storm Boris was located when it reached its maximum precipitation intensity."

- In the discussion/conclusion you argue that simulating Boris in a future climate while nudging the present-climate PV structure would yield an unrealistically strong precipitation event, so nudging frameworks should be applied with care. I agree. However, isn't that precisely the purpose of some storyline approaches—to strongly constrain dynamics in order to isolate the thermodynamic response? I see your point that a full picture of climate-change effects requires considering both dynamics and thermodynamics, but I think it would be worth clarifying how you distinguish your approach from these alternative frameworks.

We agree with the reviewer: the purpose of some storyline approaches is to strongly constrain dynamics in order to isolate the thermodynamic response. However, for this procedure to provide meaningful results, it implicitly assumes that the change in the dynamics would be small, and thus that the probability to observe the same dynamics in a warmer climate would be close to the one in the present climate. It would indeed be meaningless to constrain the dynamics strongly if the probability of observing the same dynamics is, say, 1000 times less likely in the warmer climate. Our point is precisely to criticize this assumption: we show a particular example for which assuming that the dynamics would not change leads to an overestimation of the effect of climate change. We propose to add this clarification:

"Storyline approaches --- whose goal is to isolate the thermodynamical response of a particular event by conditioning on the dynamics in different periods --- should therefore seriously consider the possibility that the dynamics constrained may be much more unlikely in future periods. This should be taken into account when assessing the practical implications of such results."

Typos:

Line 315: Fig 2a. I think is Fig. 2b

Thank you for spotting this inconsistency, now the correct reference to Fig. 2c (depicting the PV field on the 14th of September 2024) is indicated.

Line 322: reduction → difference?

We have changed as suggested, thanks.

Line 328: is 'close' referring to the target region?

Correct. 'Close' refers to a cyclone center being inside the target region, which designates the geographical region around storm Boris.

Line 333: Fig S3 I cannot identify where storms are born; is it possible to highlight cyclogenesis?

Thanks for the suggestion. We propose to add a 2D histogram to Fig. S3 to highlight the cyclone genesis locations, i.e., the first time a cyclone center is identified in the SLP field by the tracking algorithm. The update figure is displayed below (now Fig. R1.2 below). This analysis provides additional evidence that the majority of considered cyclones originate South of the Alps, which is consistent with a classic Vb track.

Line 580: 'that emerges yearly analogs' → emerges when yearly

This has been corrected, thanks.

**Links to references**

Hausfather & Glen P. Peters, 2020 <a href="https://www.nature.com/articles/d41586-020-00177-3">https://www.nature.com/articles/d41586-020-00177-3</a>
Dolores-Tesillos et al. 2022 <a href="https://wcd.copernicus.org/articles/3/429/2022/">https://wcd.copernicus.org/articles/3/429/2022/</a>

Karwat et al. 2024 <a href="https://journals.ametsoc.org/view/journals/clim/37/4/JCLI-D-23-0160.1.xml">https://journals.ametsoc.org/view/journals/clim/37/4/JCLI-D-23-0160.1.xml</a>

Ermis et al. 2024 <a href="https://iopscience.iop.org/article/10.1088/2752-5295/ad4200">https://iopscience.iop.org/article/10.1088/2752-5295/ad4200</a>

Ginesta et al. 2024

https://journals.ametsoc.org/view/journals/clim/37/21/JCLI-D-23-0761.1.xml

Messmer et al. 2015, <a href="https://doi.org/10.5194/esd-6-541-2015">https://doi.org/10.5194/esd-6-541-2015</a>

Fig. R1.2: Tracks of selected cyclones for the (a,b) seasonal and (c,d) yearly PV analogs in the CESM1 simulation in the periods (a,c) 1990–1999 and (b,d) 2091–2100. All tracks must be located within the red box at the analog time step. The red track indicates the track of storm Boris. Color shading indicates the number of cyclone track starting points within 5deg x 5deg bins.

**Reviewer 2**

The study examines Storm Boris, which caused heavy rainfall and severe flooding in central Europe. In particular, it attempts to understand how such storms may change as a result of climate change. Various methods are used, including a conditional (dynamic) method and a probabilistic method. The paper thus raises an important scientific question: to what extent does the method used to attribute climate change alter the result? It becomes clear that different processes (thermodynamic and dynamic) can have different and even opposite effects on such complex systems, as is the case for Storm Boris. The comprehensive analysis is a novelty and emphasizes that it is important to know the exact research question, but also that a comprehensive analysis is important in the attribution of climate change. The paper is well-structured and easy to follow. Therefore, I would suggest accepting this study after minor revisions.

We thank the reviewer for their positive feedback and address their comments below.

**Minor comments:**

A general comment regarding CESM. I do not understand why the authors assume that CESM would be able to generate the same precipitation intensity as the clearly higher resolved (spatially and vertically) ERA5 data. To be fair, you would need to downscale the selected analogues to a resolution that resembles the one of ERA5. Therefore, I suggest excluding the paragraph around L390 to 403 and the second row of Fig. 7 (e-h).

We understand and share the concern of the reviewer. To make a better comparison between ERA5 precipitation and CESM1 precipitation we now employ an upscaling procedure for ERA5: we use a quantile mapping transformation from the 5-d accumulated precipitation in ERA5 in August, September, October over the period 2020-2024 to the same CESM1 distribution in the present period. This is roughly equivalent to mapping the Boris storm to the most intense seasonal extreme in CESM1. We suggest updating Fig. 7 to reflect this different approach with Fig. R2.1 below, and to add the following discussion:

"None of the analogs come close to the actual event in terms of accumulated precipitation, with a difference between the most intense analog event and ERA5 of 50 mm. To make the comparison to ERA5 possible, we use a quantile mapping transformation from the 5-day accumulated precipitation distribution over the period 2000-2024 in ERA5 to the same distribution in CESM1 in the present period in the months August, September, October. Because the Boris storm is the strongest precipitation event in ERA5 during this period, this is roughly equivalent to rescaling its precipitation to the most intense seasonal extreme in CESM1."

Fig. R2.1 (replacing Fig. 7 in the manuscript): Distribution of mean precipitation in the impacted region (47.2°-51°N and 12°-19.5°E) for Boris-like events defined as analogs events (two left columns) and as maximum precipitation events (two right columns). Distribution of spatial mean 5-day accumulated precipitation over the impacted region for the (a) seasonal and (b) yearly PV analogs and the maximum (c) seasonal and (d) yearly events in the periods 1990--1999 (blue) and 2091--2100 (red). The corresponding precipitation in ERA5 (after rescaling via quantile mapping to the present CESM1 distribution) for the Storm Boris over the same region is shown with a vertical dashed line. Probability to reach the rescaled ERA5 precipitation over the impacted region for the (e) seasonal and (f) yearly PV analogs and the maximum (g) seasonal and (h) yearly events in the periods 1990--1999 (blue) and 2091--2100 (red). Probabilities are estimated using a Gamma distribution for PV analogs and a GEV distribution for maximum events. The boxplots show the distribution of estimated probabilities over bootstrapped samples (see the manuscript for more details). The black boxplots show the corresponding distribution of probability ratios.

L11: What exactly are 'Boris-like' events? The formulation "Boris-like" refers here to events that are, indeed, "like Boris", but without specifying how such a "likeness" is defined (e.g., by using seasonal or yearly large-scale flow analogs, or instead adopting an unconditional approach). As in the manuscript we test the sensitivity of the results to different event definitions, the "Boris-like" formulation allows us to encompass them more in general, regardless of the chosen definition. We will revise the manuscript so that such a formulation is used when the event definition remains unspecified: for instance, at lines 73 and 75, it would be more appropriate to talk about

"analogs of storm Boris" as this was the approach chosen by Faranda et al. (2024).

L16: ad -> at Thank you very much for spotting this typo, we have corrected it.

L17: "-," is a bit of a strange notation We adopt here the WCD policy of using en-dashes for syntactic constructions (https://www.weather-climate-dynamics.net/submission.html#english). We profit from this comment, however, to split this rather long sentence using a colon and improve readability.

L18ff: Starting at: The results obtained ... You do not specify what the results are of the unconditional method, which I think is interesting and important. I understand that the abstract is already quite long, but I think it is an important part of the study. Therefore, I suggest writing the first part of the abstract a bit more concisely, so that there is room for these results too. The sentence mentioned in the comment was actually meant to refer to the results obtained from the unconditional method. In order to be even clearer about it, we propose a slight rewording to make this more explicit: "The results obtained from the analog-based approach are then compared with an unconditional, statistics-based approach focusing only on the seasonal and yearly maxima of precipitation: the latter approach allows to recover the expected intensification of extreme precipitation in a warmer climate -- at the price, however, of considering events that do not necessarily have the same dynamics as Storm Boris."

L37f: Changing dynamics could also mean that the tracks of such Boris-like storms are less or more frequently used. We appreciate the validity of this comment for precipitation events mediated by extratropical cyclones, and we suggest to add at in this sentence the following wording "and indirectly through changes in the event frequency or seasonality". We have also highlighted cyclogenesis regions in Fig. S3, so that the origin at the lee of the Alps of cyclones involved in Boris analogs can be better appreciated. More in general, a similar conclusion to the one suggested here was actually obtained by Messmer et al. (2020), who noticed a reduction in Vb cyclone tracks due to a poleward shift of the North Atlantic storm track. However, in our analysis, we already constrain the analogs by imposing the presence of a cyclone track in a given region and, thus, partly prescribing cyclone frequency: this might complicate the assessment of possible changes in cyclone tracks in the refined analogs.

L40f: This sentence is rather long. I would recommend splitting and shortening it. This is a fair point, we suggest splitting that sentence in two and adopting a slightly more concise formulation:

"This piece of research focuses precisely on the complexity of attributing extreme precipitation events to anthropogenic global warming, moving its

steps from the case of Storm Boris (2024), a recent example of a cutoff-related, heavy precipitation event over central Europe. The effect of anthropogenic global warming on the unfolding and characteristics of the event is assessed by applying different attribution approaches to the output of a medium-resolution climate model large ensemble, with more than a thousand years of data."

L58f: The description of the path sounds more like a 'Vc' rather than a 'Vb' track. Although the initial eastward propagation of Boris' shares some similarities with Vc tracks, the cyclone considered here fulfills the Vb cyclone definition of Höfstetter and Chimani (2012) because it is located, at any point during its track, in the region between 14E–20E; 47N–60N (marked as E in Fig. R2.2b). Vc cyclones, on the other hand, fulfill all the requisites of Vb cyclones detailed in that study (i.e., origin in region O, track east of 14E, overall west-to-east track) except the passage in box E. These definitions are of course based on a subjective classification and can be opined, but such a discussion would very likely be outside the scope of this article.

Fig. R2.2: (a, left) Figure 1 of Messmer et al. (2015), depicting among others the classical tracks of Vb and Vc cyclones as defined by Van Bebber. (b, right): Figure 3 by Hofstetter and Chimani (2012) depicting the regions used for the Vb cyclone classification.

L102: European Centre for Medium Range Weather Forecasts -> ECMWF Thank you again for spotting this typo, now we use the correct abbreviation (ERA5).

L111: I am wondering if the 30 vertical levels might not be a limiting factor for your analysis. I will point out some locations later in the results again.

This question might be reformulated to inquire whether the vertical resolution of CESM1 is high enough to perform the type of analyses we propose to do. In this regard, we believe that our model has a sufficient temporal and spatial resolution. The main points in support of this statement have been mentioned in other replies, so we just recap them quickly here:

- its output has a sufficient temporal and spatial resolution to compute reliable trajectories of rapidly ascending air streams (see, e.g., Joos, Sprenger et al. 2023; we also emphasize that trajectories are computed using the wind fields on original model levels, i.e., without prior interpolation to pressure levels).
- given that the event was characterized by exceptionally strong atmospheric ascent forced by the synoptic scales, we are not concerned that resolving convective phenomena in detail would be crucial to the correct representation of precipitation (at the level of detail that is important for our study);
- the available CESM resolution is perfectly suited to apply quasi-geostrophic diagnostics.

L126: The first figure you are mentioning is Fig. 6. Generally, it is expected that the numbering starts with the first figure. Thank you for spotting this inconsistency, we have removed this mention in the revised manuscript.

L129: You are using PV at 250 hPa. What interpolation method from model to pressure levels was used for CESM data?

PV is first calculated on model levels, and then the model-level PV is linearly interpolated on the 250 hPa pressure level.

L130: What is the rationale behind choosing September 14th to represent the dynamics of Storm Boris when identifying analogues? Can you give an indication on how sensitive your results are with respect to this selection? The reason behind the choice of the 14th of September is the fact that that day featured the highest 24-hour rain accumulation over the target region in ERA5, as can be inferred from Fig. 1d (see the orange line). We did not specify that, so thank you for pointing it out.

About the second point raised, we do not expect that the choice of the 13th of September as target date would change the results (we do not consider choosing the 15th as precipitation was already declining on that day). We back up this expectation with two arguments: 1) as we consider precipitation cumulated over 5 days, moving the target day by one day could only change the amount by max 20% and probably less; 2) the analogs we would select by targeting the large-scale circulation pattern of the 13th of September would roughly match the ones we already computed for the 14th, as suggested by the lagged composite of upper-level PV in our analogs (Fig. R2.3; notice in particular that the composite at lag t-1 is very similar to the PV field of Fig. 2a in the manuscript, suggesting that the selected 14th of September analogs feature the correct evolution of the large-scale flow).

Fig. R2.3: Lagged composites of PV in the refined analogs subset, featuring an upper-level trough evolving into a PV cut-off. The PV field at lag=0 matches the one depicted in Fig. 5c.

L132: "Furthermore, for each daily mean PV field, we subtract the spatial mean of the PV field ..." This sentence is unclear to me. Do you subtract the daily spatial mean or some climatology? Thank you for the observation, we indeed subtract the daily spatial mean and we have now specified it in the revised text.

L150: The bootstrapping method is unclear to me. Do you sample a 1000 times 1050 analogues? Maybe you can rephrase this part a bit to increase understandability. Thank you for the suggestion, we suggest rephrasing for further clarity: "For every metric computed on the analogs distribution, we test the significance of the difference between the two periods using a bootstrap method. In each period, we resample inside the set of analogues, with replacement, the same number of analogues and compute the metric of interest. We repeat this resampling 1000 times to get a bootstrap distribution of the metric, and then compute the probability in this resampled distribution that the value 0 (or 1 when considering ratios) is exceeded/subceeded."

L166: You mention that you are lowering the PV anomaly threshold, but it is the same number (-0.7 PVU) as in line 163. This is unclear. Thank you for pointing this out, we suggest rephrasing it as "Since Storm Boris occurred in early fall — when PV anomalies tend to be weaker — a lower PV anomaly threshold than the original one of -1.3 PVU was applied."

L159: How can you obtain a resolution of 20 hPa in CESM with only 30 vertical levels? The spacing of 20hPa between the starting positions of the trajectories has been obtained by linear, vertical interpolation between model levels. We infer that this comment might point towards possible deficiencies in the representation of warm conveyor belt (WCB) characteristics in CESM due to the reduced vertical resolution. We are, however, convinced that WCB characteristics are properly represented in the CESM model despite the relatively small number of levels: to back up this statement, we refer to the work by Joos, Sprenger et al. (2023), and by Binder et al. (2023), who specifically targeted this issue and verified that CESM warm conveyor belts reach similar outflow heights, and are associated with a comparable vertical

mass fluxes as WCBs diagnosed from the ERA-Interim data with 60 vertical levels. We use in our analysis the same approach as in those two studies. To further clarify this point, we suggest to explicitly add to the manuscript the following sentence: "We note here that the CESM model is capable of appropriately representing WCB trajectories despite the relatively small (30) number of vertical levels, as shown in detail by Joos et al. (2023) and Binder et al. (2023)."

Binder, H., Joos, H., Sprenger, M., and Wernli, H.: Warm conveyor belts in present-day and future climate simulations – Part 2: Role of potential vorticity production for cyclone intensification, Weather Clim. Dynam., 4, 19–37, https://doi.org/10.5194/wcd-4-19-2023, 2023.

Joos, H.\*, Sprenger, M.\*, Binder, H., Beyerle, U., and Wernli, H.: Warm conveyor belts in present-day and future climate simulations – Part 1: Climatology and impacts, Weather Clim. Dynam., 4, 133–155, https://doi.org/10.5194/wcd-4-133-2023, 2023.

L189:  $1.5 \times 1.5 \rightarrow 1.5^{\circ} \times 1.5^{\circ}$  This has been corrected, thanks.

L195: I would not consider the 75th percentile as extreme precipitation. It might be strong precipitation, especially when considering the long simulation runs of CESM. This is a fair point. The choice of lowering the threshold was taken to improve sampling and have a sufficient number of events to compute meaningful statistics. To avoid using imprecise language, we reworded "extreme" to "heavy" precipitation.

Section dynamical diagnostics: It is unclear why you are applying some of the tools/algorithms to operational ECMWF data and to ERA5 reanalysis. The dynamical part of the study is, in general, based on the IFS analysis. ERA5 is used only when long-term records are needed, for instance to compute anomalies or to contextualize values with respect to climatology. This was needed only to retrieve climatological information for precipitation (to obtain the baseline distribution needed to perform quantile remapping), PV (with regards to the computation of atmospheric blocking), QG-Omega (to rank Boris with respect to other events) and the Mediterranean SST anomalies. We double-checked that using either data set does not alter the result substantially: for instance, the extremely strong QG-omega forcing for ascent during Boris ranks corresponds to the top 0.16% of the ERA5 climatology using the IFS analysis value, and to the top 0.10% when comparing ERA5 with ERA5 climatology. Differences in precipitation over the target area are also very small when comparing the two data sets. As differences between analysis and re-analysis are small at the large-scale, we are confident that such limited inconsistencies do not substantially affect the dynamical characterization of the event. We will explicitly discuss the rationale and the small differences between the two data sets in the revised version.

L231: The fact that the impact region is more over eastern Europe could imply that Boris was more like a Vc cyclone. Although we see the point of the reviewer, Boris fulfilled the definition of Vb cyclone by Höfstetter and Chimani (2012), which we take as reference. Please see the reply to the comment about line 58 above for more details.

L234: From what source/dataset do these 350 mm come from? Thank you for noticing that we did not specify the reference for this claim: we were referring to Fig. 1b, whose depicted amounts match the station-based precipitation totals depicted in the ECMWF newsletter about Storm Boris published in early 2025

(https://www.ecmwf.int/en/newsletter/182/news/severe-rain-central-europe-stor m-boris), and also to the report by the Austrian Weather Service that mentioned more than 400mm of rain recorded in St. Pölten (close to Vienna) and in other locations of the Niederösterreich province (Greilinger et al., cf. page 12).

L246: Can you identify if the uplift of the 600 hPa is triggered by dynamics or orography? We are able to do so thanks to the use of the QG diagnostic for ascent, based on the inversion of the Q-vector formulation of the omega equation (Davies 2015), which allows precisely to separate the contribution to vertical motions of synoptic-scale dynamics, and neglects other, usually more local, contributions such as the one by non-adiabatic processes or orography.

Davies, H. C., 2015: The Quasigeostrophic Omega Equation: Reappraisal, Refinements, and Relevance. *Mon. Wea. Rev.*, 143, 3–25, <a href="https://doi.org/10.1175/MWR-D-14-00098.1">https://doi.org/10.1175/MWR-D-14-00098.1</a>.

L281: If I understand Fig. 3f correctly, all of the Mediterranean has an SST anomaly of more than 1°C.

Yes, this is correct, 95% of the Mediterranean and Black Sea region experienced an SST anomaly of at least 1°C during this time period. We have tested different thresholds of SST anomalies with respect to the area occupied by them, and the result is depicted below in Fig. R2.4. While the fraction decreases for larger SST anomalies, for example 64% of the region experienced an SST anomaly of at least 2°C (see the red dots in Fig. R2.4), the fraction of moisture uptakes over an SST anomaly larger than 2°C is above 80% from 12 to 15 September 2024. This highlights that moisture uptakes occurred predominantly over regions with high SST anomalies, that cover a small(er) fraction of the Mediterranean and Black Sea. To make this clearer in the manuscript, we'll include the example of 2°C SST anomaly instead of 1° and suggest to add the following:

"Notably, over 80% of the extended Mediterranean contribution (i.e., conflating together the whole Mediterranean and Black Sea) during the first part of the event originated from areas with  $\Delta SST_{anom} > 2^{\circ}C$  relative to the 1994-2024 September climatology (Fig. 3f). Such a condition was fulfilled only by 64% of the considered area, suggesting that the elevated SSTs might have enhanced moisture uptake in the region. Further research, however, is required to determine whether the enhanced moisture uptake also resulted in an anomalously high Mediterranean contribution to precipitation totals in central Europe."

Fig. R2.4: Fraction of moisture uptakes (blue dots) over the Mediterranean and Black Sea occurring over a region with an SST anomaly above a certain threshold for moisture sources calculated for precipitation at 12 UTC on 13, 14 and 15 September 2024. The fraction of the Mediterranean and Black Sea region experiencing a mean SST anomaly from 5 to 11 September above a certain threshold is shown in red dots.

L289: What is meant by "all-year climatological distribution"? Is this based on daily means? We mean here the distribution of values registered in all seasons, including winter –the season that usually features the highest values of QG forcing for ascent. To improve clarity, we changed the wording "all-year" to "year-round" climatological distribution.

The vertical velocity (Omega) values in Figure 4a are based on IFS analysis data that are interpolated to a 0.5 x 0.5 degree latitude/longitude grid. The Omega distribution in Figure 4b, on the other hand, is based fully on ERA5 data determined from January 1980 until December 2022, with a six-hour time interval, which are consistently available also on the same grid as the IFS analysis data. In the revised manuscript, we will better specify that the climatological information is always obtained from ERA5.

L328: "... is selected if a cyclone center is located close to where Storm Boris ...". How is close defined?

Thank you for noticing this point. Indeed, 'close' refers here more specifically to a cyclone center being inside the target region, which designates the geographical region around storm Boris. The size and location of this target region are based on the box used by Messmer et al. (2015) in their Vb cyclone identification to constrain the end locations of Vb cyclones (46N–52N, 12E–19E). We shifted the location of their original box slightly northeast (45N–53N, 16E–25E) to better capture the track of storm Boris, which –despite

fulfilling the Vb criterion- propagated unusually eastward compared to a textbook Vb cyclone. To simplify the analysis and interpretation, we also dropped the requirement that the cyclone tracks must end in the box, and instead just require that a cyclone center must be present in the box at the time step of the analogue.

We also suggest adding the following clarification: "Therefore, we refine the selection of the PV analogs in each period by checking for the presence of a surface cyclone in the region where Boris propagated through. This target region is based on the Vb cyclone track end location constraint used by Messmer at al. 2015, and shifted northeastward to account for the track of storm Boris (45N-53N, 16E-25E; see Fig. S3)."

L332: South -> south This has been corrected, thanks.

L335: Does "propagate into the target region from the North Atlantic" mean that the track passes by the Alps only on the northern side only? Thanks for suggesting this clarification, that we believe will help the interpretation of Fig. S3. We suggest a rephrased sentence incorporating "...a larger fraction of them propagates into the target region from the North Atlantic compared to the seasonal PV analogs, remaining north of the Alps."

L338ff: I do not understand this sentence. We reworded this and the following sentence to clarify our aim in performing the analysis of euclidean distance and of Pearson correlation, that are discussed in the rest of the paragraph. It would now read: "In order to quantitatively corroborate the visual impression of Fig. 5, we additionally compute over the same region where the analogs are determined (32°-70°N, 5°W-40°E) the euclidean distance between PV and SLP fields in the analogs and the ones of the event, and the Spearman spatial correlation between the analog precipitation field and the one of the event in ERA5 (regridded conservatively at the model resolution)."

L349: Also, the fact that Vb cyclones are only characterised by the track of their surface position justifies this refinement. Thanks for the suggestion, we integrated it in the text as follows: "On average, events with a low SLP distance tend to have a better precipitation correlation, lending further support to the proposed analog refinement procedure based on the position of the surface cyclone."

L369f: "the seasonal analogs do not display the same drying as the mean" This sentence is not very clear, as it could indicate that the drying pattern just looks different, but in fact, there is almost no drying visible. Thank you for noticing this imprecise formulation, we rephrased it to the more specific "...the seasonal analogs do not display significant changes,...'

L374ff: Could the small fraction of WCB detected also be related to the comparably low vertical resolution of CESM with only 30 levels? As specified in the reply to a previous comment, WCB characteristics should be properly represented in the CESM model despite the relatively small number of levels (as shown in Joos et al. 2023, Binder et al. 2023). Specifically targeting WCB frequency, Joos et al. (2023), noticed in their Sec. 3 that "Comparing the WCB climatology in ERA-Interim (Fig. 2a and c) to the one calculated based on HIST (Fig. 2b and d), it is striking to see the similarity between the two, which thus points to CESM's capability to realistically simulate WCBs. In fact, both the frequency amplitude and the geographical location of their inflow, ascent, and outflow regions agree very well between both data sets."

L383f: This sentence is difficult to understand. First, you mention seasonal and yearly analogues, then you refer to the warming of the yearly and then the seasonal, but the numbers in the brackets are again for seasonal and then yearly. I suggest reorganising this sentence to make it more intuitive. Thank you for the suggestion, we suggest rewording the sentence so that first the seasonal, and then the yearly analogs are consistently mentioned in this order. The sentence would read now: "This leads to a different 2-m air temperature warming response for seasonal and yearly PV analogs, with the former warming twice as rapidly as the latter (+7.2K vs +3.6K)."

L386: I suggest using the word accumulated precipitation instead of cumulated precipitation. You will have to replace this on multiple occasions.

Thank you for the recommendation, we will modify the text as suggested.

L388f: Why do you expect the same precipitation intensity between ERA5 and CESM? The resolution of these two data sets is quite substantially different, making it very hard for CESM to actually reach the same intensity as in ERA5. This is even true if you would regrid ERA5 to the CESM grid, as the underlying resolution is still much higher. We believe that this point has been addressed in the reply to the first comment, thanks to the upscaling of precipitation based on quantile mapping.

L389ff: "The mean precipitation decreases" with respect to what? We modified the sentence for further clarity to indicate that the change is with respect to the present conditions: "The mean precipitation expected in the future model climate from Boris-like events decreases for seasonal analogs (-5.6 mm, p<0.001) while it does not change significantly when considering yearly analogs (+2.5 mm, p=0.06)".

L391 to the end of the section: It is unclear to me why you are assuming that CESM should be able to capture the same intensity as ERA5. Therefore, I do not see the point of this analysis here. We believe that this point has also been addressed in

the reply to the first comment, thanks to the upscaling of precipitation based on quantile mapping.

L395: There is 'estimate', 'estimation' and 'estimate' in one sentence. Please rephrase. Thank you for the suggestion, we rephrase it to "To evaluate the sensitivity of the fit with respect to limited sample size, we estimate..."

L415: Again, I think part of this is due to the resolution of CESM. It would be interesting to know how precipitation intensity increases for these events if downscaled to the same resolution as ERA5, which, of course, is beyond the scope of this study. Indeed, this would be interesting: however, we believe the upscaling to the upscaling of precipitation based on quantile mapping (introduced to reply to the first comment) could already give some insight in a relatively inexpensive way.

L445: Why are you looking at the 90th percentile here and not at the 75th percentile as mentioned in the method section?

The 90th percentile mentioned here is the percentile of the conditional distribution, i.e. the distribution of the analogues. The 75th percentile mentioned in the method section refers to the climatological quantile. This has been precised in the text.

L519ff: This sentence is unclear to me. Thank you for this comment, we took the occasion to make this long sentence more concise and we avoided the "Boris-like" formulation: "The example of Storm Boris shows it clearly, because the large-scale circulation analogs do not necessarily correspond to the events of maximum precipitation in the model climatology."

L545: I agree that CESM has a high temporal resolution with 6-hourly output, but the spatial and especially the vertical resolution is not really high. We agree with the reviewer that the use of "high" might be debatable, and change the formulation to "relatively high resolution".

L580: "..., that emerges yearly analogs are considered." This part does not seem to fit into this sentence. This has been corrected as follows: "...that emerges when yearly analogs are considered".

Figure 2: The difference between the blue and purple contours is difficult to see, at least in a printed version. Thank you for noticing it. Stippling has been added to the area delimited by the blue contour to highlight the presence of atmospheric blocking and further distinguish it from the purple 2PVU contour.

Why is PV shown at 320 K and not at 250 hPa, as it was introduced in the method section?

Potential vorticity is materially conserved on isentropic surfaces (e.g., Hoskins et al. 1986) and, thus, is usually depicted on them. As the interpolation to isentropic surface would have been an unnecessary hurdle in the model (which features indeed more than 1000 years of 6-hourly data), we chose to keep it simple and use the more readily available PV field at 250hPa.

Supplementary Figures: Make sure that they appear in an increasing order in the text. I think S16 is mentioned before S15 and S14. Thank you for spotting this inconsistency, we will make sure that the ordering is correct in the reviser version.

**Links to references**

Binder et al. 2023, <a href="https://doi.org/10.5194/wcd-4-19-2023">https://doi.org/10.5194/wcd-4-19-2023</a>

Davies 2015: https://doi.org/10.1175/MWR-D-14-00098.1

Hoskins et al 1985. https://doi.org/10.1002/gj.49711147002

Joos\*, Sprenger\*, et al. 2023, <a href="https://doi.org/10.5194/wcd-4-133-2023">https://doi.org/10.5194/wcd-4-133-2023</a>

Messmer et al. 2015, https://doi.org/10.5194/esd-6-541-2015

Messmer et al. 2020, https://doi.org/10.1080/16000870.2020.1724021

---

## Author Response (AR1)

We would like to thank the two reviewers for the constructive feedback, because addressing their comments allowed us to improve both scientific consistency and presentation clarity.

The main changes brought to the revised manuscript are:
- A change in the probability ratios in Fig. 7, that are now computed with respect to an upscaled value of the 5-day accumulated precipitation in ERA5;
- A clarification of when IFS analysis data and Reanalysis data are used in the study, with slightly updated Fig. 1 and Fig. 4;
- An improved structuring of the Discussion section (Sect. 5), with the addition of a separate "Limitations" sub-section (Sect. 5.4);
- We reworded some sentences following reviewer's suggestions and corrected some typos (e.g., in the caption of Fig. 4, forcing for ascent by upper levels "above 650hPa" was changed with the correct "above 550hPa" to match the already correct description in the Sec. 2.3).

**Reviewer 1**

This study performs a process-understanding projection of Storm Boris, with a very interesting approach that combines climate and weather information. The paper also shows the potential of a methodology to disentangle the dynamic and thermodynamic signals of climate change on specific events and assess their contributions separately. This stands in contrast to other methodologies that either ignore the dynamical information of the event or constrain it so tightly that part of the signal is lost. I very much enjoyed reading the paper; the structure and language are very clear and easy to follow.

We thank the reviewer for their positive evaluation of our work and address their comments below.

A concern relates to the model performance in simulating mesoscale extreme events within this type of cyclones. Other studies using the same model (Dolores-Tesillos et al. 2023; Karwat et al. 2024; and from some of the authors themselves Binder et al. 2024; Joos et al. 2024) have raised questions in this regard. I understand that, as the authors state, this work can be seen as a "meta-attribution"* study rather than a definitive attribution analysis. However, I think it is important to discuss whether CESM is suitable for simulating this type of storm. My understanding is that the model does not reproduce weak cyclones very well, especially in the Mediterranean. Could this limitation influence your results?

It is true that, as noted by the reviewer and given its coarse resolution, we do not expect that the CESM1 model used here reproduces correctly the climatology of mesoscale extreme precipitation events. However, we have shown that for the Boris storm a large part of the forcing for ascent comes from large-scale, quasi-geostrophic (QG) forcing. This mechanism is resolved at the resolution of the model and therefore we can reasonably expect the

model to represent correctly how this would change in a warmer climate. Moreover, even if the model cannot reproduce the climatology of weak cyclones in the Mediterranean or cut-off lows in Central Europe, we expect this bias to cancel out because we compare two periods inside the same model and, therefore, to obtain a reasonable estimate of the change between the two periods. This is especially true when conditioning on similar atmospheric patterns as the one observed during the Boris storm: while the model may be biased in how it represents the frequency of cut-offs and Vb cyclones, we may also expect that its physics is reasonably close to the truth *conditional* on the observation of a cut-off and a Vb cyclone. To address this point, we added a new subsection "Limitations" to Sect. 5 and explicitly discuss these limitations at lines 571-582:

"The CESM1 climate model used here to evaluate changes in Boris-like storms has a relatively coarse resolution and its capacity to simulate mesoscale extreme events can therefore be questioned, as already pointed out by previous works (Dolores-Tesillos et al. 2022, Joos et al. 2023, Karwat et al 2024, Binder and Wernli 2025). We do not expect the model to reproduce correctly the climatology of mesoscale extreme precipitation events, nor the intensity of extreme precipitation (see our discussion of Fig. 7). However, we have shown that, for the Boris storm, a large part of the forcing for ascent comes from the large-scale forcing of the PV cutoff. This mechanism is resolved at the resolution of the model and therefore we can reasonably expect that its evolution in a future climate with weaker cutoffs is well reproduced. Moreover, even if the model cannot reproduce the climatology of weak cyclones in the Mediterranean or cut-off lows in Central Europe, because we compare two periods inside the same model, we expect the bias with respect to the real world climatology to cancel out to give a reasonable estimate of the *change* between the two periods. This is especially true when conditioning on similar atmospheric patterns: the model may be biased in how it represents the frequency of cut-offs and Vb cyclones but we may also expect that its physics is reasonably close to the truth *conditional* on the observation of a cut-off and a Vb cyclone."

Also, more generally, what are the limitations of using a single model, and how do the authors justify using a high-end, unrealistic, rcp scenario (Hausfather & Peters 2020)?

We agree that the RCP5-8.5 scenario is a high-end, economically unrealistic warming scenario. However, by imposing a high-warming scenario to the climate system, we also expect to improve the signal to noise ratio when detecting dynamical changes in weather systems, which are expected to be much smaller than the thermodynamical changes. Using a single-model is indeed also a major limitation – especially when it comes to investigating

**highly-parametrized precipitation – but few models have available 1000 years of 3D data at 6 hourly resolution as the one we are using here for our analogs and precipitation scaling analysis. We now explicitly discuss these limitations in the new subsection "Limitations" added to Sec. 5 (lines 583-588).**

I understand that the authors apply a methodology within the framework of extreme event attribution and therefore repeatedly frame the study in terms of "attribution." However, in my view this may be somewhat misleading for communication purposes, and in several places, I would suggest rephrasing in terms of "projection." In the attribution literature, "counterfactual" typically refers to a climate that could have occurred (without anthropogenic forcing) or that might occur under different forcing conditions, while "factual" refers to the present climate. For projections, the concept of "future counterfactuals" is also used (e.g., Ermis et al. 2024). It is therefore confusing to use "counterfactual" for the present climate and "factual" for the future climate (e.g., lines 136–138). A clarification of terminology here would be very helpful.

**We agree with the reviewer that this may be confusing, despite the fact that the separation between factual and counterfactual periods is a priori arbitrary. As suggested, we now use in the manuscript the term 'future counterfactual' for the simulations with the future RCP.**

Other points that would benefit from justification or clarification:

- Choice of the domain. It seems quite large; why do you need to include the main PV reservoir over Scandinavia? As you state in the discussion, the domain can strongly influence results. Have you performed sensitivity tests, e.g. focusing only on the cut-off low?

**We agree that the domain chosen to find analogs is quite large, including the PV reservoir over Scandinavia. We chose this domain because, as argued in section 3, the Boris event is characterized by the detachment of a PV cut-off situated between two large anticyclones, whose growth and motion was partly responsible for the thinning and the successive breaking of a PV streamer. The domain choice was thus influenced by the dynamical contextualization performed in section 3, and more specifically by the need of appropriately resolving such high-latitude anticyclones. We emphasize, however, that the analog refinement procedure based on the position of the low-level cyclone strongly filters the analogues found, so that changing marginally the size of this large domain has few consequences on the results. We also tried to use a much smaller domain centered on the cut-off and found similar results for precipitation (see Figure R1.1 below), especially the reduced intensity of the cutoff, but the analogs featured a strongly anticyclonically tilted PV streamer**

**and did not feature the broad anticyclone over western Russia and, thus, did not adequately represent the large-scale dynamics during the event.**

[Figure]

*Figure R1.1: in the first two columns, as in Figure 5 of the manuscript, are shown precipitation and 250hPa PV composites for analogs computed over a smaller region centered on the cut-off location (red box in sub-plots a,b). The first line depicts analogs in present climate, second line in future climate and third line their difference. In the third and fourth column, for comparison, analogs are computed over the same large region used in the study. Here we show all 1050 analogs with no refinement.*

- I understand (and like) the use of PV as the first step to define the event. However, as the authors state, most analog studies use slp or geopotential height, and storm Boris had a peculiar track of the surface cyclone. Have you considered, for comparison, first identifying analogues of SLP (or Vb tracks, as in Ginesta et al. 2024, using the same dataset) and then constraining them with a cut-off low based on PV? Do you think this would yield more/better (or fewer/worse) analogues, or perhaps a stronger precipitation correlation?

**The methodology proposed by the reviewer would essentially be the reverse of the one we employed: selecting analogues of low-level cyclones first and then filtering those associated with mid-level PV cut-offs. This is indeed something that is feasible in general, and we do expect that it would give similar results as the one used here (because the intersection of two sets is independent of**

which set is considered first). However, for this particular case we think it may present some difficulties, mainly because we are working in a region with strong topography where the use of the SLP to define low-level cyclones could be problematic (see our discussion at lines 175-178 on the tracking of the low-level cyclone for Boris) and lead to spurious analogues. In Ginesta et al. 2024 the low-level cyclone occurred over the sea or close to the coast, where the use of SLP is better justified. These elements have been now detailed in lines 368-371:

"Note that the reverse procedure could also be implemented, first selecting on SLP analogs and in a second step refining the selection with events featuring an upper-level PV cutoff. We expect that the two procedures would give similar results: however, given that the complex orography of the impacted region might lead to spurious SLP patterns, basing the initial selection on upper tropospheric features seems a preferable option."

- Am I correct in assuming that you compare with Z500 analogues here because the SLP signal of this cyclone is too weak for a meaningful comparison?

Yes, this is indeed one of the arguments. Another reason is that the cut-off signal would not be seen on SLP whereas it is clearly seen on Z500 (and even more visible in the PV field), and, as discussed above, we start the procedure by finding analogues of the cut-off low.

- Can you justify the choice of the domain of target/analogue slp identification (red box)?

The target region for identifying the low-level cyclone is not a domain where we compute SLP analogues, but rather where we impose the center of the low-level cyclone to be located. The size and location of the box are based on the regions used by Messmer et al. (2015) in their Vb cyclone identification to constrain the end locations of Vb cyclones (46–52N, 12–19E). We shifted the location of their original box slightly northeast (45N–53N, 16E–25E) to better capture the track of storm Boris, which propagated unusually eastward compared to a textbook Vb cyclone. To simplify the analysis and interpretation with respect to Messmer et al. (2015), we also dropped the requirement that the cyclone tracks must end in the box, and instead just require that a cyclone center must be present in the box at the time step of the analogue.
We added the following clarification at lines 340-341:
"Therefore, we refine the selection of the PV analogs in each period by checking for the presence of a surface cyclone in the region where Boris propagated through. This target region is based on the Vb cyclone track end constraint used by Messmer at al. 2015, slightly shifted northeastward to account for the track of storm Boris (45N–53N, 16E–25E; see Fig. S3)".

**Furthermore, we have noticed while replying to this comment that we were using the wording "target region" to define both the impacted region (red box in Fig. 1) and the region in which we are expecting the cyclone to be located to select the refined analogs. To avoid misunderstandings, we now distinguish between the "impacted region" (introduced at lines 244-245) and the "target region" for the cyclones.**

- Line 328: should analogues also be defined at the time of maximum precipitation intensity, or is this criterion applied only for Boris itself?

**Thank you for pointing that out, this sentence can be misleading indeed: the box to filter low-level cyclone is defined with respect to where the center of the low-level cyclone was at the day of maximum precipitation during the Boris storm, but we do not impose that analogues need to have the center of their low-level cyclone in this box AND the day of their maximum precipitation at the same moment. This has been precised at lines 341-342: "Thus, an analog is selected if a cyclone center is located close to where Storm Boris was located when it reached its maximum precipitation intensity"**

- In the discussion/conclusion you argue that simulating Boris in a future climate while nudging the present-climate PV structure would yield an unrealistically strong precipitation event, so nudging frameworks should be applied with care. I agree. However, isn't that precisely the purpose of some storyline approaches—to strongly constrain dynamics in order to isolate the thermodynamic response? I see your point that a full picture of climate-change effects requires considering both dynamics and thermodynamics, but I think it would be worth clarifying how you distinguish your approach from these alternative frameworks.

**We agree with the reviewer: the purpose of some storyline approaches is to strongly constrain dynamics in order to isolate the thermodynamic response. However, for this procedure to provide meaningful results, it implicitly assumes that the change in the dynamics would be small, and thus that the probability to observe the same dynamics in a warmer climate would be close to the one in the present climate. It would indeed be meaningless to constrain the dynamics strongly if the probability of observing the same dynamics is, say, 1000 times less likely in the warmer climate. Our point is precisely to criticize this assumption: we show a particular example for which assuming that the dynamics would not change leads to an overestimation of the effect of climate change. We further clarified this aspect by adding the following at lines 563-566:**

**"Storyline approaches --- whose goal is to isolate the thermodynamical response of a particular event by conditioning on the dynamics in different periods --- should therefore seriously consider the possibility that the**

dynamics constrained may be much more unlikely in future periods. This should be taken into account when assessing the practical implications of such results."

Typos:

Line 315: Fig 2a. I think is Fig. 2b

Thank you for spotting this inconsistency, now the correct reference to Fig. 2c (depicting the PV field on the 14th of September 2024) is indicated.

Line 322: reduction → difference?

We have changed as suggested, thanks.

Line 328: is 'close' referring to the target region?

Correct. 'Close' refers to a cyclone center being inside the target region, which designates the geographical region around storm Boris. This has been made explicit now in lines 338-340.

Line 333: Fig S3 I cannot identify where storms are born; is it possible to highlight cyclogenesis?

Thanks for the suggestion. We added a 2D histogram to Fig. S3 which shows the cyclone genesis locations, i.e., the first time a cyclone center is identified in the SLP field by the algorithm. The updated figure is displayed below (Fig. R1.2). This analysis provides additional evidence that the majority of considered cyclones originate South of the Alps, which is consistent with a classic Vb track.

[Figure]

*Fig. R1.2 (now Figure S3): Tracks of selected cyclones for the (a,b) seasonal and (c,d) yearly PV analogs in the CESM1 simulation in the periods (a,c) 1990–1999 and (b,d) 2091–2100. All tracks must be located within the red box at the analog time step. The red track indicates the track of storm Boris. Color shading indicates the number of cyclone track starting points within 5deg x 5deg bins.*

Line 580: 'that emerges yearly analogs' → emerges when yearly

**This has been corrected, thanks.**

**Links to references**

Hausfather & Glen P. Peters, 2020
https://www.nature.com/articles/d41586-020-00177-3
Dolores-Tesillos et al. 2022 https://wcd.copernicus.org/articles/3/429/2022/
Karwat et al. 2024
https://journals.ametsoc.org/view/journals/clim/37/4/JCLI-D-23-0160.1.xml
Ermis et al. 2024 https://iopscience.iop.org/article/10.1088/2752-5295/ad4200
Ginesta et al. 2024
https://journals.ametsoc.org/view/journals/clim/37/21/JCLI-D-23-0761.1.xml
Messmer et al. 2015, https://doi.org/10.5194/esd-6-541-2015

**Reviewer 2**

The study examines Storm Boris, which caused heavy rainfall and severe flooding in central Europe. In particular, it attempts to understand how such storms may change as a result of climate change. Various methods are used, including a conditional (dynamic) method and a probabilistic method. The paper thus raises an important scientific question: to what extent does the method used to attribute climate change alter the result? It becomes clear that different processes (thermodynamic and dynamic) can have different and even opposite effects on such complex systems, as is the case for Storm Boris. The comprehensive analysis is a novelty and emphasizes that it is important to know the exact research question, but also that a comprehensive analysis is important in the attribution of climate change. The paper is well-structured and easy to follow. Therefore, I would suggest accepting this study after minor revisions.

We thank the reviewer for their positive feedback and address their comments below.

Minor comments:

A general comment regarding CESM. I do not understand why the authors assume that CESM would be able to generate the same precipitation intensity as the clearly higher resolved (spatially and vertically) ERA5 data. To be fair, you would need to downscale the selected analogues to a resolution that resembles the one of ERA5. Therefore, I suggest excluding the paragraph around L390 to 403 and the second row of Fig. 7 (e-h).

We understand and share the concern of the reviewer. To make a better comparison between ERA5 precipitation and CESM1 precipitation we now employ an upscaling procedure for ERA5: we use a quantile mapping transformation from the 5-d accumulated precipitation in ERA5 in August, September, October over the period 2020-2024 to the same CESM1 distribution in the present period. This is roughly equivalent to mapping the Boris storm to the most intense seasonal extreme in CESM1. This is now detailed in lines 406-411:

"None of the analogs come close to the actual event in terms of accumulated precipitation, with a difference between the most intense analog event and ERA5 of 50 mm. To make the comparison to ERA5 possible, we use a quantile mapping transformation from the 5-day accumulated precipitation distribution over the period 2000-2024 in ERA5 to the same distribution in CESM1 in the present period in the months August, September, October. Because the Boris storm is the strongest precipitation event in ERA5 during this period, this is roughly equivalent to rescaling its precipitation to the most intense seasonal extreme in CESM1."

L11: What exactly are 'Boris-like' events? **The formulation "Boris-like" refers here to events that are, indeed, "like Boris", but without specifying how such a "likeness" is defined (e.g., by using seasonal or yearly large-scale flow analogs, or instead adopting an unconditional approach). As in the manuscript we test the sensitivity of the results to different event definitions, the "Boris-like" formulation allows us to encompass them more in general, regardless of the chosen definition. We have revised the manuscript so that such a formulation is used when the event definition remains unspecified: for instance, at lines 73 and 75, we changed "Boris-like events" to "analogs of storm Boris" as this was the chosen methodology by Faranda et al. (2024).**

L16: ad -> at **Thank you very much for spotting this typo, we have corrected it.**

L17: "–," is a bit of a strange notation **We adopt here the WCD policy of using en-dashes for syntactic constructions (https://www.weather-climate-dynamics.net/submission.html#english). We profit from this comment, however, to split this rather long sentence using a colon and improve readability.**

L18ff: Starting at: The results obtained … You do not specify what the results are of the unconditional method, which I think is interesting and important. I understand that the abstract is already quite long, but I think it is an important part of the study. Therefore, I suggest writing the first part of the abstract a bit more concisely, so that there is room for these results too. **The sentence mentioned in the comment was actually meant to refer to the results obtained from the unconditional method. In order to be even clearer about it, we propose a slight rewording of lines 18-21 to make this more explicit: "The results obtained from the analog-based approach are then compared with an unconditional, statistics-based approach focusing only on the seasonal and yearly maxima of precipitation: the latter approach allows to recover the expected intensification of extreme precipitation in a warmer climate -- at the price, however, of considering events that do not necessarily have the same dynamics as Storm Boris."**

L37f: Changing dynamics could also mean that the tracks of such Boris-like storms are less or more frequently used. **We appreciate the validity of this comment for precipitation events mediated by extratropical cyclones, and we thus added to the abstract at line 17 the following wording "and indirectly through changes in the event frequency or seasonality". We have also highlighted cyclogenesis regions in Fig. S3, so that the origin at the lee of the Alps of cyclones involved in Boris analogs can be better appreciated. More in general, a similar conclusion to the one suggested here was actually obtained by Messmer et al. (2020), who noticed a reduction in Vb cyclone tracks due to a poleward shift of the North Atlantic storm track. However, in our analysis, we already constrain the analogs by imposing the presence of a cyclone track in a given region and,**

thus, partly prescribing cyclone frequency: this might complicate the assessment of possible changes in cyclone tracks in the refined analogs.

L40f: This sentence is rather long. I would recommend splitting and shortening it. **This is a fair point, we suggest splitting that sentence in two and adopting a slightly more concise formulation (lines 41-45):**
**"This piece of research focuses precisely on the complexity of attributing extreme precipitation events to anthropogenic global warming, moving its steps from the case of Storm Boris (2024), a recent example of a cutoff-related, heavy precipitation event over central Europe. The effect of anthropogenic global warming on the unfolding and characteristics of the event is assessed by applying different attribution approaches to the output of a medium-resolution climate model large ensemble, with more than a thousand years of data."**

L58f: The description of the path sounds more like a 'Vc' rather than a 'Vb' track. **Although the initial eastward propagation of Boris' shares some similarities with Vc tracks, the cyclone considered here fulfills the Vb cyclone definition of Höfstetter and Chimani (2012) because it is located, at any point during its track, in the region between 14E–20E; 47N–60N (marked as E in Fig. R1b). Vc cyclones, on the other hand, fulfill all the requisites of Vb cyclones detailed in that study (i.e., origin in region O, track east of 14E, overall west-to-east track) except the passage in box E. These definitions are of course based on a subjective classification and can be opined, but such a discussion would very likely be outside the scope of this article.**

[Figure]

**Figure 1.** Trajectories of the barometric minima between 1876 – 1880, as defined by W.J. van Bebber in 1981. The trajectory that defines Vb-cyclones is highlighted in black.

[Figure]

**Figure 3:** Study region for tracking analysis (2.25°W–23.63°E and 37.57°N–58.88°N) and areas for the categorisation of Vb-like 700 hPa cyclone tracks: "O" stands for the area of origin for Vb- and Vcd-tracks, "E" ..exit area for Vb, dashed gray line at 14°East for Vb and Vcd, "R"...core study region for the detection of all remaining tracks.

*Fig. R2.1: (left) Figure 1 of Messmer et al. (2015), depicting among others the classical tracks of Vb and Vc cyclones as defined by Van Bebber. (right): Figure 3 by Hofstetter and Chimani (2012) depicting the regions used for the Vb cyclone classification.*

L102: European Centre for Medium Range Weather Forecasts -> ECMWF **Thank you again for spotting this typo, now we use the correct abbreviation (ERA5).**

L111: I am wondering if the 30 vertical levels might not be a limiting factor for your analysis. I will point out some locations later in the results again.

**This question might be reformulated to inquire whether the vertical resolution of CESM1 is high *enough* to perform the type of analyses we propose to do. In this regard, we believe that our model has a sufficient temporal and spatial resolution. The main points in support of this statement have been mentioned in other replies, so we just recap them quickly here:**

- **its output has a sufficient temporal and spatial resolution to compute reliable trajectories of rapidly ascending air streams (see, e.g., Joos, Sprenger et al. 2023; we also emphasize that trajectories are computed using the wind fields on original model levels, i.e., without prior interpolation to pressure levels).**
- **given that the event was characterized by exceptionally strong atmospheric ascent forced by the synoptic scales, we are not concerned that resolving convective phenomena in detail would be crucial to the correct representation of precipitation (at the level of detail that is important for our study);**
- **the available CESM resolution is perfectly suited to apply quasi-geostrophic diagnostics.**

L126: The first figure you are mentioning is Fig. 6. Generally, it is expected that the numbering starts with the first figure. **Thank you for spotting this inconsistency, we have removed this mention in the revised manuscript.**

L129: You are using PV at 250 hPa. What interpolation method from model to pressure levels was used for CESM data?
**PV is first calculated on model levels, and then the model-level PV is linearly interpolated on the 250 hPa pressure level.**

L130: What is the rationale behind choosing September 14th to represent the dynamics of Storm Boris when identifying analogues? Can you give an indication on how sensitive your results are with respect to this selection? **The reason behind the choice of the 14th of September is the fact that that day featured the highest 24-hour rain accumulation over the target region in ERA5, as can be inferred from Fig. 1 (see the orange line). We did not specify that, so thank you for pointing it out: this rationale is now made explicit at lines 133-135.**
**About the second point raised, we do not expect that the choice of the 13th of September as target date would change the results (we do not consider choosing the 15th as precipitation was already declining on that day). We back up this expectation with two arguments: 1) as we consider precipitation cumulated over 5 days, moving the target day by one day could only change the amount by max 20% and probably less; 2) the analogs we would select by targeting the large-scale circulation pattern of the 13th of September would**

roughly match the ones we already computed for the 14th, as suggested by the lagged composite of upper-level PV in our analogs (Fig. R2.2; notice in particular that the composite at lag t-1 is very similar to the PV field of Fig. 2a in the manuscript, suggesting that the selected 14th of September analogs feature the correct evolution of the large-scale flow).

[Figure]

*Fig. R2.3: Lagged composites of PV in the refined analogs subset, featuring an upper-level trough evolving into a PV cut-off. The PV field at lag=0 matches the one depicted in Fig. 5c.*

L132: "Furthermore, for each daily mean PV field, we subtract the spatial mean of the PV field …" This sentence is unclear to me. Do you subtract the daily spatial mean or some climatology? **Thank you for the observation, we indeed subtract the daily spatial mean and we have now specified it in the revised text.**

L150: The bootstrapping method is unclear to me. Do you sample a 1000 times 1050 analogues? Maybe you can rephrase this part a bit to increase understandability. **Thank you for the suggestion, we have rephrased these sentences for further clarity (lines 153-157): "For every metric computed on the analogs distribution, we test the significance of the difference between the two periods using a bootstrap method. In each period, we resample inside the set of analogues, with replacement, the same number of analogues and compute the metric of interest. We repeat this resampling 1000 times to get a bootstrap distribution of the metric, and then compute the probability in this resampled distribution that the value 0 (or 1 when considering ratios) is exceeded/subceeded."**

L166: You mention that you are lowering the PV anomaly threshold, but it is the same number (-0.7 PVU) as in line 163. This is unclear. **Thank you for pointing this out, we have rephrased it as "Since Storm Boris occurred in early fall — when PV anomalies tend to be weaker — a lower PV anomaly threshold than the original one of -1.3 PVU was applied." (line 173).**

L159: How can you obtain a resolution of 20 hPa in CESM with only 30 vertical levels? **The spacing of 20hPa between the starting positions of the trajectories has been obtained by linear, vertical interpolation between model levels. We infer that this comment might point towards possible deficiencies in the representation of warm conveyor belt (WCB) characteristics in CESM due to the reduced vertical resolution. We are, however, convinced that WCB**

characteristics are properly represented in the CESM model despite the relatively small number of levels: to back up this statement, we refer to the work by Joos, Sprenger et al. (2023), and by Binder et al. (2023), who specifically targeted this issue and verified that CESM warm conveyor belts reach similar outflow heights, and are associated with a comparable vertical mass fluxes as WCBs diagnosed from the ERA-Interim data with 60 vertical levels. We use in our analysis the same approach as in those two studies. To further clarify this point, we added to the manuscript the following sentence at lines 165-167: "We note here that the CESM model is capable of appropriately representing WCB trajectories despite the relatively small (30) number of vertical levels, as shown in detail by Joos et al. (2023) and Binder et al. (2023)."

Binder, H., Joos, H., Sprenger, M., and Wernli, H.: Warm conveyor belts in present-day and future climate simulations – Part 2: Role of potential vorticity production for cyclone intensification, Weather Clim. Dynam., 4, 19–37, https://doi.org/10.5194/wcd-4-19-2023, 2023.
Joos, H.*, Sprenger, M.*, Binder, H., Beyerle, U., and Wernli, H.: Warm conveyor belts in present-day and future climate simulations – Part 1: Climatology and impacts, Weather Clim. Dynam., 4, 133–155, https://doi.org/10.5194/wcd-4-133-2023, 2023.

L189: 1.5 x 1.5 -> 1.5° x 1.5° This has been corrected, thanks.

L195: I would not consider the 75th percentile as extreme precipitation. It might be strong precipitation, especially when considering the long simulation runs of CESM. This is a fair point. The choice of lowering the threshold was taken to improve sampling and have a sufficient number of events to compute meaningful statistics. To avoid using imprecise language, we reworded "extreme" to "heavy" precipitation.

Section dynamical diagnostics: It is unclear why you are applying some of the tools/algorithms to operational ECMWF data and to ERA5 reanalysis. The dynamical part of the study is, in general, based on the IFS analysis. ERA5 is used only when long-term records are needed, for instance to compute anomalies or to contextualize values with respect to climatology. This was needed only to retrieve climatological information for precipitation (to obtain the baseline distribution needed to perform quantile remapping), PV (with regards to the computation of atmospheric blocking), QG-Omega (to rank Boris with respect to other events) and the Mediterranean SST anomalies. We double-checked that using either data set does not alter the result substantially: for instance, the extremely strong QG-omega forcing for ascent during Boris ranks corresponds to the top 0.16% of the ERA5 climatology using the IFS analysis value, and to the top 0.10% when comparing ERA5 with

**ERA5 climatology (this has been now specified at lines 299-302). Similarly, differences in precipitation over the target area are also very small when comparing the two data sets, as can be inferred from the updated Fig. 1. As differences between analysis and re-analysis are small at the large-scale, we are confident that such limited inconsistencies do not substantially affect the dynamical characterization of the event.**

L231: The fact that the impact region is more over eastern Europe could imply that Boris was more like a Vc cyclone. **Although we see the point of the reviewer, Boris fulfilled the definition of Vb cyclone by Höfstetter and Chimani (2012), which we take as reference. Please see the reply to the comment about line 58 above for more details.**

L234: From what source/dataset do these 350 mm come from? **Thank you for noticing that we did not specify the reference for this claim: we were referring to Fig. 1b, whose depicted amounts match the station-based precipitation totals depicted in the ECMWF newsletter about Storm Boris published in early 2025 (https://www.ecmwf.int/en/newsletter/182/news/severe-rain-central-europe-storm-boris), and also to the report by the Austrian Weather Service that mentioned more than 400mm of rain recorded in St. Pölten (close to Vienna) and in other locations of the Niederösterreich province (Greilinger et al., cf. page 12). We added those missing references at lines 241-243 and changed "350mm" to "400mm".**

L246: Can you identify if the uplift of the 600 hPa is triggered by dynamics or orography? **We are able to do so thanks to the use of the QG diagnostic for ascent, based on the inversion of the Q-vector formulation of the omega equation (Davies et al. 2015), which allows precisely to separate the contribution to vertical motions of synoptic-scale dynamics, and neglects other, usually more local, contributions such as the one by non-adiabatic processes or orography.**

**Davies, H. C., 2015: The Quasigeostrophic Omega Equation: Reappraisal, Refinements, and Relevance.** *Mon. Wea. Rev.*, **143, 3–25, https://doi.org/10.1175/MWR-D-14-00098.1.**

L281: If I understand Fig. 3f correctly, all of the Mediterranean has an SST anomaly of more than 1°C.
**Yes, this is correct, 95% of the Mediterranean and Black Sea region experienced an SST anomaly of at least 1°C during this time period. We have tested different thresholds of SST anomalies with respect to the area occupied by them, and the result is depicted below in Fig. R2.4. While the fraction decreases for larger SST anomalies, for example 64% of the region**

experienced an SST anomaly of at least 2°C (see the red dots in Fig. R2.4), the fraction of moisture uptakes over an SST anomaly larger than 2°C is above 80% from 12 to 15 September 2024. This highlights that moisture uptakes occurred predominantly over regions with high SST anomalies, that cover a small(er) fraction of the Mediterranean and Black Sea. To make this clearer in the manuscript, we'll include the example of 2°C SST anomaly instead of 1° and modified the related discussion (lines 288-294):

"Notably, over 80% of the extended Mediterranean contribution (i.e., conflating together the whole Mediterranean and Black Sea) during the first part of the event originated from areas with $\Delta SST_{anom}$ >2°C relative to the 1994-2024 September climatology (Fig. 3f). Such a condition was fulfilled only by 64% of the considered area, suggesting that the elevated SSTs might have enhanced moisture uptake in the region. Further research, however, is required to determine whether the enhanced moisture uptake also resulted in an anomalously high Mediterranean contribution to precipitation totals in central Europe."

[Figure]

*Fig. R2.3: Fraction of moisture uptakes (blue dots) over the Mediterranean and Black Sea occurring over a region with an SST anomaly above a certain threshold for moisture sources calculated for precipitation at 12 UTC on 13, 14 and 15 September 2024. The fraction of the Mediterranean and Black Sea region experiencing a mean SST anomaly from 5 to 11 September above a certain threshold is shown in red dots.*

L289: What is meant by "all-year climatological distribution"? Is this based on daily means? We mean here the distribution of values registered in all seasons, including winter –the season that usually features the highest values of QG forcing for ascent. To improve clarity, we changed the wording "all-year" to "year-round" climatological distribution.

The vertical velocity (Omega) values in Figure 4a are based on IFS analysis data that are interpolated to a 0.5 x 0.5 degree latitude/longitude grid. The Omega distribution in Figure 4b, on the other hand, is based fully on ERA5 data determined from January 1980 until December 2022, with a six-hour time interval, which are consistently available also on the same grid as the IFS analysis data. In the revised manuscript, we now specify that climatological information is always obtained from ERA5 (lines 103-104).

L328: "… is selected if a cyclone center is located close to where Storm Boris …". How is close defined?

**Thank you for noticing this point. Indeed, 'close' refers here more specifically to a cyclone center being inside the target region, which designates the geographical region around storm Boris. The size and location of this target region are based on the box used by Messmer et al. (2015) in their Vb cyclone identification to constrain the end locations of Vb cyclones (46N–52N, 12E–19E). We shifted the location of their original box slightly northeast (45N–53N, 16E–25E) to better capture the track of storm Boris, which –despite fulfilling the Vb criterion– propagated unusually eastward compared to a textbook Vb cyclone. To simplify the analysis and interpretation, we also dropped the requirement that the cyclone tracks must end in the box, and instead just require that a cyclone center must be present in the box at the time step of the analogue.**
**We added the following clarification at lines 338-341: "Therefore, we refine the selection of the PV analogs in each period by checking for the presence of a surface cyclone in the region where Boris propagated through. This target region is based on the Vb cyclone track end location constraint used by Messmer at al. 2015, and shifted northeastward to account for the track of storm Boris (45°N–53°N, 16°E–25°E; see Fig. S3)."**

L332: South -> south **This has been corrected, thanks.**

L335: Does "propagate into the target region from the North Atlantic" mean that the track passes by the Alps only on the northern side only? **Thanks for requesting this clarification, that we believe will help the interpretation of Fig. S3. We rephrased the sentence incorporating "...a larger fraction of them propagates into the target region from the North Atlantic compared to the seasonal PV analogs, remaining north of the Alps." (lines 347-349).**

L338ff: I do not understand this sentence. **We reworded this and the following sentence to clarify our aim in performing the analysis of euclidean distance and of Pearson correlation, that are discussed in the rest of the paragraph. It now reads: "In order to quantitatively corroborate the visual impression of Fig. 5, we additionally compute over the same region where the analogs are determined (32°N-70°N, 5°W-40°E) the euclidean distance between PV and SLP fields in the analogs and the ones of the event, and the Spearman spatial correlation between the analog precipitation field and the one of the event in ERA5 (regridded conservatively at the model resolution)." (lines 351-355).**

L349: Also, the fact that Vb cyclones are only characterised by the track of their surface position justifies this refinement. **Thanks for the suggestion, we integrated**

**it in the text as follows: "On average, events with a low SLP distance tend to have a better precipitation correlation, lending further support to the proposed analog refinement procedure based on the position of the surface cyclone." (lines 362-364).**

L369f: "the seasonal analogs do not display the same drying as the mean" This sentence is not very clear, as it could indicate that the drying pattern just looks different, but in fact, there is almost no drying visible. **Thank you for noticing this imprecise formulation, we rephrased it to the more specific "...the seasonal analogs do not display significant changes,..." (line 387).**

L374ff: Could the small fraction of WCB detected also be related to the comparably low vertical resolution of CESM with only 30 levels? **As specified in the reply to a previous comment, WCB characteristics should be properly represented in the CESM model despite the relatively small number of levels (as shown in Joos et al. 2023, Binder et al. 2023). Specifically targeting WCB frequency, Joos et al. (2023), noticed in their Sec. 3 that "Comparing the WCB climatology in ERA-Interim (Fig. 2a and c) to the one calculated based on HIST (Fig. 2b and d), it is striking to see the similarity between the two, which thus points to CESM's capability to realistically simulate WCBs. In fact, both the frequency amplitude and the geographical location of their inflow, ascent, and outflow regions agree very well between both data sets."**

L383f: This sentence is difficult to understand. First, you mention seasonal and yearly analogues, then you refer to the warming of the yearly and then the seasonal, but the numbers in the brackets are again for seasonal and then yearly. I suggest reorganising this sentence to make it more intuitive. **Thank you for the suggestion, we have reworded the sentence so that first the seasonal, and then the yearly analogs are consistently mentioned in this order. The sentence reads now: "This leads to a different 2-m air temperature warming response for seasonal and yearly PV analogs, with the former warming twice as rapidly as the latter (+7.2K vs +3.6K, see Fig. S14)." (lines 400-402).**

L386: I suggest using the word accumulated precipitation instead of cumulated precipitation. You will have to replace this on multiple occasions.

**Thank you for the recommendation, we have modified the text as suggested.**

L388f: Why do you expect the same precipitation intensity between ERA5 and CESM? The resolution of these two data sets is quite substantially different, making it very hard for CESM to actually reach the same intensity as in ERA5. This is even true if you would regrid ERA5 to the CESM grid, as the underlying resolution is still much higher. **We believe that this point has been addressed in the reply to the**

**first comment, thanks to the upscaling of precipitation based on quantile mapping.**

L389ff: "The mean precipitation decreases" with respect to what? **We modified the sentence for further clarity to indicate that the change is with respect to the present conditions: "The mean precipitation expected in the future model climate from Boris-like events decreases for seasonal analogs (-5.6 mm, p<0.001) while it does not change significantly when considering yearly analogs (+2.5 mm, p=0.06)".**

L391 to the end of the section: It is unclear to me why you are assuming that CESM should be able to capture the same intensity as ERA5. Therefore, I do not see the point of this analysis here. **We believe that this point has been addressed in the reply to the first comment, thanks to the rescaling of precipitation based on quantile mapping.**

L395: There is 'estimate', 'estimation' and 'estimate' in one sentence. Please rephrase. **Thank you for the suggestion, we rephrase it to "To evaluate the sensitivity of the fit with respect to limited sample size, we estimate…" (lines 417-418).**

L415: Again, I think part of this is due to the resolution of CESM. It would be interesting to know how precipitation intensity increases for these events if downscaled to the same resolution as ERA5, which, of course, is beyond the scope of this study. **Indeed, this would be very interesting, ideally with new simulations at a higher resolution.**

L445: Why are you looking at the 90th percentile here and not at the 75th percentile as mentioned in the method section?
**The 90th percentile mentioned here is the percentile of the conditional distribution, i.e. the distribution of the analogues. This has been specified in the text (lines 467-468). The 75th percentile mentioned in the method section, on the other hand, refers to the climatological quantile.**

L519ff: This sentence is unclear to me. **Thank you for this comment, we took the occasion to make this long sentence more concise and we avoided the "Boris-like" formulation: "The example of Storm Boris shows it clearly, because the large-scale circulation analogs do not necessarily correspond to the events of maximum precipitation in the model climatology." (lines 542-544).**

L545: I agree that CESM has a high temporal resolution with 6-hourly output, but the spatial and especially the vertical resolution is not really high. **We agree with the reviewer that the use of "high" might be debatable, and change the formulation to "relatively high resolution".**

L580: "..., that emerges yearly analogs are considered." This part does not seem to fit into this sentence. **Thank you for spotting the typo, it has been corrected by adding the missing word "when": "...that emerges when yearly analogs are considered".**

Figure 2: The difference between the blue and purple contours is difficult to see, at least in a printed version. **Thank you for noticing it. Stippling has been added to the area delimited by the blue contour to highlight the presence of atmospheric blocking and further distinguish it from the purple 2PVU contour.**

Why is PV shown at 320 K and not at 250 hPa, as it was introduced in the method section?
**Potential vorticity is materially conserved on isentropic surfaces (e.g., Hoskins et al. 1985) and, thus, is usually depicted on them. As the interpolation to isentropic surface would have been an unnecessary hurdle in the model (which features indeed more than 1000 years of 6-hourly data), we chose to keep it simple and use the more readily available PV field at 250hPa.**

Supplementary Figures: Make sure that they appear in an increasing order in the text. I think S16 is mentioned before S15 and S14. **Thank you for spotting this inconsistency, now the former Fig. S16 has become S14 and the numbering has been re-adjusted.**

**Links to references**

Binder et al. 2023, https://doi.org/10.5194/wcd-4-19-2023
Davies 2015: https://doi.org/10.1175/MWR-D-14-00098.1
Hoskins et al 1985. https://doi.org/10.1002/qj.49711147002
Joos*, Sprenger*, et al. 2023, https://doi.org/10.5194/wcd-4-133-2023
Messmer et al. 2015, https://doi.org/10.5194/esd-6-541-2015
Messmer et al. 2020, https://doi.org/10.1080/16000870.2020.1724021

---

## Author Response (AR2)

**We would like to thank again the two reviewers for their time and their constructive assessment of the manuscript. We concisely reply below to the additional suggestions of Reviewer 1.**

**Reviewer 1**

I am happy with the revised manuscript. I just have a couple of small comments:

The paper is already pretty long. Section 2.3 Dynamical Diagnostic doesn't feel essential in the main text and could be moved to the supplementary. (e.g. extreme precipitation objects aren't used in the main text and are only mentioned once to point to the supplementary material) If readers want the details of how these tools are computed, they can look there.

**Thank you for this recommendation. We agree with the reviewer's point and have moved to the Supplement the description of heavy precipitation objects. However, given that all other diagnostics were shown in at least one Figure of the main manuscript, and that cyclone tracking and QG-omega diagnostics are very important for the arguments made in Sect. 4, we have decided to keep the descriptions of the other diagnostics in the main text.**

In Section 3.5, maybe rethink the word "Summary" as it could be confusing, or even adding something like summary of the dynamical features or similar might work better.

**Thanks for the suggestion, to avoid confusion we changed the title of Subsection 3.5 to "Summary: Salient dynamical features of Storm Boris", and modified the title of Sec. 3 to "Meteorological analysis of Storm Boris".**